# A Non-vacuous Test Error Guarantee for Deep Learning without Altering the Model

## Abstract

Deep neural networks (NN) with millions or billions of parameters can perform really well on unseen data, after being trained from a finite training set. Various prior theories have been developed to explain such excellent ability of NNs, but do not provide a meaningful bound on the test error. Some recent theories are non-vacuous under some stringent assumptions and extensive modification (e.g. compression, quantization) to the trained model of interest. Therefore, those prior theories provide a guarantee for the modified models only. In this paper, we present two novel bounds on the true error of a model. One of our bounds can be exactly computable from the training set only, without altering the model, and hence provides a theoretical guarantee for a trained model. Our approach is to decompose the data space into different local areas to approximate the local errors by using training samples in a controlled way, then use those local errors to approximate the true error of a model. Our bounds are verified on 32 modern NNs, which were trained by Pytorch on the ImageNet dataset. The exactly computable bound is found to be non-vacuous. To the best of our knowledge, this is the first non-vacuous bound at this large scale (NNs with more than 600M parameters, ImageNet), without altering those 32 trained models.

## 1 Introduction

Deep neural networks (NNs) have been enabling many breakthroughs (Silver et al., 2016; Jumper et al., 2021; Achiam et al., 2023) in different areas. However, there remains a big gap between theory and practice of modern NNs. In particular, it is largely unclear (Zhang et al., 2021) about *Why can deep NNs generalize well on unseen data after being trained from a finite number of samples?* This question relates to the generalization ability of a trained model. The standard learning theories suffer from various difficulties to provide a reasonable explanation. Various approaches have been studied, e.g. Radermacher complexity (Bartlett et al., 2017), algorithmic stability (Brutzkus & Globerson, 2021), algorithmic robustness (Sokolić et al., 2017), PAC-Bayes (Biggs & Guedj, 2022).

Some recent theories (Zhou et al., 2019; Lotfi et al., 2024a;b) are really promising, as they can provide meaningful bounds on the test error of some models. Dziugaite & Roy (2017) obtained a non-vacuous bound by optimizing a distribution over NN parameters. (Zhou et al., 2019; Mustafa et al., 2024) bounded the expected error of a *stochastic NN* by using off-the-shelf compression methods. Those theories follow the PAC-Bayes approach. On the other hand, Nadjahi et al. (2024) showed the potential of the stability-based approach. Although making a significant progress, those theories are meaningful for small and *stochastic NNs* only. Lotfi et al. (2024a;b) made a significant step to analyze the generalization ability of big/huge NNs, such as large language models (LLM). Using state-of-the-art quantization, finetuning and some other techniques, the PAC-Bayes bounds by (Lotfi et al., 2024b;a) are non-vacuous for huge LLMs, e.g., GPT-2 and LLamMA2. Those bounds significantly push the frontier of deep learning theory.

**Can we obtain a non-vacuous true error bound for a specific/trained neural network without modifying that model?** The true (expected) error tells how well a model can generalize on unseen data, and hence can explain the performance of a trained model. Bounding the true error is a fundamental problem in learning theory (Mohri et al., 2018), but arguably challenging for mordern NNs. Many prior theories (Zhou et al., 2019; Lotfi et al., 2022; Nadjahi et al., 2024) were developed for *stochastic models*, but not for a trained model of interest. Lotfi et al. (2024a;b) made a significant

Table 1: Recent approaches for analyzing generalization error. ✓ means "Required" or "Yes". The upper part shows the required assumptions about differrent aspects, e.g., hypothesis space, loss function, training or finetuning. The lower part reports non-vacuousness in different situations.

| Approach | Weight norm | Alg. Stability | Alg. Robustness | Mutual Info | PAC-Bayes | Ours |
|---|---|---|---|---|---|---|
| **Requirement:** | | | | | | |
| Model compressibility | | | | ✓ | ✓ | ✓ |
| Train or finetune | | | | ✓ | ✓ | ✓ |
| Lipschitz loss | ✓ | ✓ | | ✓ | | |
| *Finite* hypothesis space | | | | | | ✓ |
| **Non-vacuousness** for: | | | | | | |
| *Stochastic* models only | | ✓ | | ✓ | ✓ | |
| Trained models | | | | | ✓ | ✓ |
| Training size $> 1$ M | | | | | ✓ | ✓ |
| Model size $> 600$ M | | | | | ✓ | ✓ |

progress to remove "stochasticity". For example, Lotfi et al. (2024b) provided a non-vacuous bound for the 2-bit quantized (and finetuned) versions of LLamMA2. Nonetheless, those theories require to use a method for intensively quantizing or compressing the trained model. This means that those theories are for the quantized or compressed models, and *hence may not necessarily be true for the original (unquantized or uncompressed) models.*[1] This is a major limitation of those bounds. Such a limitation calls for novel theories that directly work with a given model.

Our contributions in this work are as follows:

- We develop a novel bound on the expected error of a trained model $h$. Unlike prior bounds, our result does not require stringent assumptions. It jointly encodes the complexity of the data distribution and the local behavior of $h$ in different regions of the data space. The main technical challenge lies in approximating an intractable term that summarizes the true error of $h$ across local regions. We resolve this by analyzing fine-grained properties of small and binomial random variables.
- We further derive a tractable bound that can be computed directly from the training set, without any modification to $h$. This bound preserves the structural advantages of the general bound while being practically applicable. It can provide a (computable) theoretical guarantee for the test error of a trained model. These properties overcome the key limitations of prior theories. A detailed comparison with existing approaches is provided in Table 1.
- We conduct extensive experiments on 32 modern neural networks which were pretrained by Pytorch on the ImageNet dataset. The results consistently yield non-vacuous upper bounds, even under conservative parameter settings. To the best of our knowledge, this is the first demonstration of non-vacuous theoretical guarantees at such large scale (spanning CNNs, Transformers, and over 600M-parameter models) without altering the trained models.

*Organization:* The next section presents a comprehensive survey about related work. We then present our novel bounds in Section 3, accompanied with more detailed comparisons. Section 4 contains our empirical evaluation for some pretrained NNs. Section 5 concludes the paper. Proofs and more experimental details can be found in appendices.

## 2 RELATED WORK

Various approaches have been studied to analyze generalization capability, e.g., Radermacher complexity, algorithmic stability, algorithmic robustness, Mutual-infomation based bounds, PAC-Bayes. Those approaches connect different aspects of a learning algorithm or hypothesis to generalization.

---

[1]A modified version $h'$ can have an entirely different generalization behavior from the original model $h$, especially when significant compression or quantization is used to obtain $h'$ of small size as evidenced with the quality of GPT-2 in (Lotfi et al., 2024a). More importantly, the *gap* between the expected losses $F(P, h)$ (to be defined in Section 3) and $F(P, h')$ of the two models is often unknown in prior works. Such an unknown suggests that a good bound for $F(P, h')$ does not necessarily translate into a good bound for $F(P, h)$. A heavier compresion can cause the gap to be larger, suggesting a (potentially) significant tradeoff.

**Norm-based bounds** (Bartlett et al., 2017; Golowich et al., 2020; Galanti et al., 2023b; Graf et al., 2022) is one of the earliest approaches to understand NNs. The existing studies often use Rademacher complexity to provide data- and model-dependent bounds on the generalization error. An NN with smaller weight norms will have a smaller bound, suggesting better generalization on unseen data. Nonetheless, the norms of weight matrices are often large for practical NNs (Arora et al., 2018). Therefore, most existing norm-based bounds are vacuous.

**Algorithmic stability** (Bousquet & Elisseeff, 2002; Shalev-Shwartz et al., 2010; Charles & Papailiopoulos, 2018; Kuzborskij & Lampert, 2018) is an approach to studying a learning algorithm. Basically, those theories suggest that a more stable algorithm can generalize better. Stable algorithms are less likely to overfit the training set, leading to more reliable predictions. The stability requirement in those theories is that a replacement of one sample for the training set will not significantly change the loss of the trained model. Such an assumption is really strong. One drawback is that achieving stability often requires restricting model complexity, potentially sacrificing predictive accuracy on challenging datasets. Therefore, this approach has a limited success in understanding deep NNs.

**Algorithmic robustness** (Xu & Mannor, 2012; Sokolić et al., 2017; Kawaguchi et al., 2022; Than et al., 2025) is a framework to study generalization capability. It says that a robust learning algorithm can produce robust models which can generalize well on unseen data. This approach provides another lens to understand a learning algorithm and a trained model. However, it requires the assumption that the learning algorithm is robust, i.e., the loss of the trained model changes little in the small areas around the training samples. Such an assumption is really strong and cannot apply well for modern NNs, since many practical NNs suffer from adversarial attacks (Madry et al., 2018; Zhou et al., 2022). Than et al. (2025) showed that those theories are often vacuous.

**Neural Tangent Kernel** (Jacot et al., 2018; Arora et al., 2019) provides a theoretical lens to study generalization of NNs by linking them to kernel methods in the infinite-width limit. As networks grow wider, their training dynamics under gradient descent can be approximated by a kernel function which remains constant throughout training. This perspective simplifies the analysis of complex neural architectures. The framework enables explicit generalization bounds, and a deeper understanding of how NN architecture and initialization affect learning. However, the main limitation of this framework comes from its assumptions, such as the *infinite-width* regime and fixed kernel during training, may not fully capture the behavior of finite, practical NNs. Some other studies (Lee et al., 2022) can remove the infinite-width regime but assume the *infinite depth*.

**Mutual information (MI)** (Xu & Raginsky, 2017; Feldman & Vondrak, 2019; Nadjahi et al., 2024) has emerged as a powerful tool for analyzing generalization by quantifying the dependency between a model's learned representations and the data. Since a trained model contains the (compressed) knowledge learned from the training samples, MI offers a principled framework for studying the trade-off between compression and predictive accuracy. However, the existing MI-based theories (Xu & Raginsky, 2017; Wang et al., 2021; Sefidgaran et al., 2022; Nadjahi et al., 2024) have a notable drawback: computing MI in high-dimensional, non-linear settings is computationally challenging. This drawback poses significant challenges for analyzing deep NNs, although (Nadjahi et al., 2024) obtained some promising results on small NNs.

**PAC-Bayes** (McAllester, 1999; Haddouche & Guedj, 2023; Biggs & Guedj, 2023; Awasthi et al., 2020; Pérez-Ortiz et al., 2021) recently has received a great attention, and provide non-vacuous bounds (Zhou et al., 2019; Mustafa et al., 2024) for some NNs. Those bounds often estimate $\mathbb{E}_{\hat{h}}[F(P, \hat{h})]$ which is the expectation of the test error over the posterior distribution of $\hat{h}$. It means that those bounds are for a *stochastic model* $\hat{h}$. Hence they provide limited understanding for a specific deterministic model $h$. Neyshabur et al. (2018) provided an attempt to derandomization for PAC-Bayes but resulted in vacuous bounds for modern neural networks (Arora et al., 2018). Some recent attempts to derandomization include (Viallard et al., 2024; Clerico et al., 2025).

**Non-vacuous bounds for NNs:** Dziugaite & Roy (2017) obtained a non-vacuous bound for NNs by finding a posterior distribution over neural network parameters that minimizes the PAC-Bayes bound. Their optimized bound is non-vacuous for a stochastic MLP with 3 layers trained on MNIST dataset. Zhou et al. (2019) bounded the population loss of a stochastic NNs by using compressibility level of a NN. Using off-the-shelf neural network compression schemes, they provided the first non-vacuous bound for LeNet-5 and MobileNet, trained on ImageNet with more than 1.2M samples.

Lotfi et al. (2022) developed a compression method to further optimize the PAC-Bayes bound, and estimated the error rate of 40.9% for MobileViT on ImageNet. Mustafa et al. (2024) provided a non-vacuous PAC-Bayes bound for adversarial population loss for VGG on CIFAR10 dataset. Galanti et al. (2023a) presented a PAC-Bayes bound which is non-vacuous for Convolutional NNs with up to 20 layers and for CIFAR10 and MNIST. Akinwande et al. (2024) provided a non-vacuous PAC-Bayes bound for prompts. Although making a significant progress for NNs, those bounds are non-vacuous for stochastic neural networks only. Biggs & Guedj (2022) provided PAC-Bayes bounds for deterministic models and obtain (empirically) non-vacuous bounds for a specific class of (SHEL) NNs with a single hidden layer, trained on MNIST and Fashion-MNIST. Nonetheless, it is unclear about how well those bounds apply to bigger or deeper NNs.

Towards understanding big/huge NNs, Lotfi et al. (2024a;b) made a significant step that provides non-vacuous bounds for LLMs. While the PAC-Bayes bound in (Lotfi et al., 2024a) can work with LLMs trained from i.i.d data, the recent bound in (Lotfi et al., 2024b) considers token-level loss for LLMs and applies to dependent settings, which is close to the practice of training LLMs. Using both model quantization, finetuning and some other techniques, the PAC-Bayes bound by (Lotfi et al., 2024b) is shown to be non-vacuous for huge LLMs, e.g., LLamMA2. Those bounds significantly push the frontier of learning theory towards building a solid foundation for DL.

Nonetheless, there are two main drawbacks of those bounds (Lotfi et al., 2024a;b). First, model quantization or compression is required in order to obtain a good bound. It means, those bounds are for the quantized or compressed models, and *hence may not necessarily be true for the original (unquantized or uncompressed) models*. For example, (Lotfi et al., 2024b) provided a non-vacuous bound for the 2-bit quantized versions of LLamMA2, instead of their original pretrained versions. Second, those bounds require the assumption that *the model (hypothesis) family is finite*, meaning that a learning algorithm only searches in a space with finite number of specific models. Although such an assumption is reasonable for the current computer architectures, those bounds cannot explain a trained model that belongs to families with infinite (or uncountable) number of members, which are provably prevalent. In contrast, our bounds apply directly to any specific model without requiring any modification or support. A comparison between our bounds and prior approaches about some key aspects is presented in Table 1.

## 3 ERROR BOUNDS

In this section, we present novel bounds for the error of a given model. The first bound provides a general form which depends on the complexity of the data distribution and the trained model. This bound cannot be exactly computed, but serves as the theoretical foundation. The last bound provides an explicit error estimate, which can be computed directly from any given dataset.

*Notations:* $S$ often denotes a dataset and $|S|$ denotes its size/cardinality. $\Gamma$ denotes a partition of the data space. $[K]$ denotes the set $\{1, ..., K\}$ of natural numbers at most $K$. $\ell$ denotes a loss function, and $h$ often denotes a model or hypothesis of interest.

Consider a hypothesis (or model) $h : \mathcal{X} \to \mathcal{Y}$ which maps from an input space $\mathcal{X}$ to an output space $\mathcal{Y}$, and a loss function $\ell : (h, (x, y)) \mapsto l \in \mathbb{R}$, where $(x, y) \in \mathcal{X} \times \mathcal{Y}$. Each $\ell(h, z)$ tells the loss (or quality) of $h$ at an instance $z \in \mathcal{Z} := \mathcal{X} \times \mathcal{Y}$. Given a distribution $P$ defined on $\mathcal{Z}$, the quality of $h$ is measured by its *expected loss* $F(P, h) = \mathbb{E}_{z \sim P}[\ell(h, z)]$. Quantity $F(P, h)$ tells the generalization ability of model $h$; a smaller $F(P, h)$ implies better generalization on unseen data.

For analyzing generalization ability, we are often interested in estimating (or bounding) $F(P, h)$. Sometimes this expected loss is compared with the *empirical loss* of $h$ on a data set $S = \{z_1, ..., z_n\} \subseteq \mathcal{Z}$, which is defined as $F(S, h) = \frac{1}{n} \sum_{z \in S} \ell(h, z)$. Note that a small $F(S, h)$ does not neccessarily imply good generalization of $h$, since overfitting may appear. Therefore, our ultimate goal is to estimate $F(P, h)$ directly.

Let $\Gamma(\mathcal{Z}) := \bigcup_{i=1}^{K} \mathcal{Z}_i$ be a partition of $\mathcal{Z}$ into $K$ disjoint nonempty subsets. Denote $S_i = S \cap \mathcal{Z}_i$, and $n_i = |S_i|$ as the number of samples falling into $\mathcal{Z}_i$, meaning that $n = \sum_{j=1}^{K} n_j$. Let $T = \{i \in [K] : n_i > 0\}$ contain the indices of areas for which some samples of $S$ appear, $a_i(h) = \mathbb{E}_z[\ell(h, z)|z \in \mathcal{Z}_i]$ as the expected (local) loss of $h$ in area $\mathcal{Z}_i$ for each $i \in [K]$, and $a_o = \max_{j \notin T} a_j(h)$.

## 3.1 General bounds

The first result incorporates some properties of the data distribution and the trained model.

**Theorem 3.1.** *Given a partition $\Gamma$ and a bounded nonnegative loss $\ell$, consider a model $\boldsymbol{h}$ which may depend on a dataset $\boldsymbol{S}$ with $n$ i.i.d. samples from distribution $P$. Denote $p_i = \Pr_{\boldsymbol{z} \sim P}(\boldsymbol{z} \in \mathcal{Z}_i)$ as the probability measure of area $\mathcal{Z}_i$ for $i \in [K]$, and $u = \sum_{i=1}^{K} \gamma n p_i (1 + \gamma n p_i)$. For any constants $\gamma \geq 1$, $\delta_1 \geq \exp(-\frac{u \ln \gamma}{4n-3})$ and $\delta_2 > 0$, we have the following with probability at least $1 - \delta_1 - \delta_2$:*

$$F(P, \boldsymbol{h}) \leq F(\boldsymbol{S}, \boldsymbol{h}) + C\sqrt{\frac{u}{2n^2} \ln \frac{1}{\delta_1}} + g(\Gamma, \boldsymbol{h}, \delta_2) \tag{1}$$

*where $g(\Gamma, \boldsymbol{h}, \delta_2) = \frac{\sqrt{\ln(2K/\delta_2)}}{n} \sum_{i \in \boldsymbol{T}} \sqrt{n_i} \left(a_o + \sqrt{2} a_i(\boldsymbol{h})\right) + \frac{2\ln(2K/\delta_2)}{n}(a_o|\boldsymbol{T}| + \sum_{i \in \boldsymbol{T}} a_i(\boldsymbol{h}))$ and $C = \sup_{\boldsymbol{z} \in \mathcal{Z}} \ell(\boldsymbol{h}, \boldsymbol{z})$.*

This theorem suggests that the expected loss cannot be far from the empirical loss $F(\boldsymbol{S}, \boldsymbol{h})$. The gap between the two is at most $C\sqrt{\frac{u}{2n^2} \ln \frac{1}{\delta_1}} + g(\Gamma, \boldsymbol{h}, \delta_2)$. Such a gap represents the uncertainty of our bound and mostly depends on the sample size $n$, the trained model $\boldsymbol{h}$, the data distribution $P$ and the partition $\Gamma$. We emphasize that bound (1) has some interesting properties:

- *First, it does not require any assumption about the hypothesis family and learning algorithm.* This is an advantage over many approaches including algorithmic stability (Li et al., 2024), robustness (Xu & Mannor, 2012; Kawaguchi et al., 2022), Radermacher complexity (Bartlett et al., 2017; Galanti et al., 2023b). This bound focuses directly on the the model $\boldsymbol{h}$ of interest, helping it to be tighter than many prior bounds.
- *Second, it depends on the complexity of the data distribution.* Note that $u$ encodes the complexity of $P$. For a uniform partition $\Gamma$, a more structured distribution $P$ can have a higher sum $\sum_{i=1}^{K} p_i^2$. As an example of structured distributions, a Gaussian with a small variance has the most probability density in a small area around its mean and lead to a high $p_i$ for some $i$. Meanwhile a less structured distribution (e.g. uniform) can produce a small $\sum_{i=1}^{K} p_i^2$ and hence smaller $u$. Furthermore, the constant $|\boldsymbol{T}|$ also provides "shallow" infomation about the distribution complexity, e.g., a uniform $P$ can result in a large $|\boldsymbol{T}|$. To the best of our knowledge, such an explicit dependence on the distribution complexity is rare in prior theories.
- *Third, it is model-dependent.* Some particular properties of model $\boldsymbol{h}$ are encoded in both $g(\Gamma, \boldsymbol{h}, \delta_2)$ and the empirical loss . A better model $\boldsymbol{h}$ will lead to smaller $a_i$'s and hence $g$. On the contrary, a worse model can have a bigger $g$, leading to a higher RHS of (1).

It is worth noticing the similarity between our bound (1) and robustness-based bounds in (Kawaguchi et al., 2022; Than et al., 2025). $F(\boldsymbol{S}, \boldsymbol{h}) + g(\Gamma, \boldsymbol{h}, \delta_2)$ is the common part in those bounds. Our bound (1) contains $C\sqrt{\frac{u}{2n^2} \ln \frac{1}{\delta_1}}$ that encodes the complexity of the data distribution, whereas the bounds in (Kawaguchi et al., 2022; Than et al., 2025) use a robustness quantity that measures the sensitivity of the loss w.r.t. a change in the input. While prior bounds are not amenable to be exactly computed from a training set, our bound enables to easily derive a computable and non-vacuous bound (below). This is the main advantage of bound (1).

One limitation of bound (1) is that it is not diminishing as $n$ increases while fixing the partition size $K$. It can be seen from the second term, i.e., $C\sqrt{0.5(\frac{\gamma}{n} + \gamma^2 \sum_{i=1}^{K} p_i^2) \ln \frac{1}{\delta_1}}$. Luckily, this issue can be easily fixed by allowing $K$ to increase with $n$, owing to the following result.

**Corollary 1.** *Given the notations in Theorem 3.1, consider a continuous distribution $P$ supported on a convex domain $\mathcal{Z}$. For any $K > 0, \delta_2 > 0$, $\delta_1 \geq \exp\left(-(\gamma n + \gamma^2 n^2/K)(\ln \gamma)/(4n - 3)\right)$, with probability at least $1 - \delta_1 - \delta_2$, we have: $F(P, \boldsymbol{h}) \leq F(\boldsymbol{S}, \boldsymbol{h}) + C\sqrt{0.5\left(\frac{\gamma}{n} + \frac{\gamma^2}{K}\right)\ln \frac{1}{\delta_1}} + g(\Gamma, \boldsymbol{h}, \delta_2)$.*

*Remark* 1 (Convergence rate). This result suggests that by choosing $K = O(n^\beta)$ for $\beta \in [0, 1]$, the test error is bounded by $F(\boldsymbol{S}, \boldsymbol{h}) + O(n^{-\beta/2})$. Note that there is a tradeoff between $K$ and $g$. A large $K$ can potentially produce a large $g$, as evidenced in our ablation later on real data. Therefore a balanced choice for $K$ seems to be $O(n^{1/2})$, making our bound scale as $O(n^{-1/4})$. Note that this convergence rate seems to be sub-optimal, and hence leaves open room for future improvement.

*Proof sketch for Theorem 3.1.* The detailed proof appears in Appendix A. We focus on bounding the probability $\Pr\left(F(P, \boldsymbol{h}) - F(\boldsymbol{S}, \boldsymbol{h}) \geq \phi\right)$, for some gap $\phi$. Note that $F(P, \boldsymbol{h}) - F(\boldsymbol{S}, \boldsymbol{h}) = A + B$, where $A = F(P, \boldsymbol{h}) - \sum_i \frac{n_i}{n} a_i(\boldsymbol{h})$ and $B = \sum_i \frac{n_i}{n} a_i(\boldsymbol{h}) - F(\boldsymbol{S}, \boldsymbol{h})$. Therefore, our proof estimates $\Pr(A \geq g)$ and $\Pr(B \geq t)$ for some constant $t$. Once they are known, we can use the union bound to obtain a bound on $\Pr\left(F(P, \boldsymbol{h}) - F(\boldsymbol{S}, \boldsymbol{h}) \geq g + t\right)$ as desired. We use a result from (Kawaguchi et al., 2022) to bound $\Pr(A \geq g)$. The remaining task is to estimate $\Pr(B \geq t)$, which is **the main challenge**. This challenge requires approximating an intractable quantity from a data set.

We resolve this challenge by developing Theorem A.1. Its proof contains three main steps:

1. First we show $\Pr(B \geq t) \leq e^{-yt} \mathbb{E}_{\boldsymbol{h}, \boldsymbol{n}} \left[ \mathbb{E}_{\boldsymbol{S}} \left[ e^{yB} | \boldsymbol{h}, \boldsymbol{n} \right] \right]$, for $\boldsymbol{n} = \{n_1, ..., n_K\}$ and some $y$.

2. We next estimate $\mathbb{E}_{\boldsymbol{S}} \left[ e^{yB_K} | \boldsymbol{h}, \boldsymbol{n} \right]$. Overall, we make sure that $\mathbb{E}_{\boldsymbol{S}} \left[ e^{yB} | \boldsymbol{h}, \boldsymbol{n} \right] \leq e^{\psi(y, \boldsymbol{n})}$, for some function $\psi(y, \boldsymbol{n})$ which does not depend on $\boldsymbol{h}$. As a result $\Pr(B \geq t) \leq e^{-yt} \mathbb{E}_{\boldsymbol{n}} e^{\psi(y, \boldsymbol{n})}$.

3. The last step is to bound $\mathbb{E}_{\boldsymbol{n}} e^{\psi(y, \boldsymbol{n})}$. This requires us to develop various analyses for small random variables in Appendix B. A suitable choice for $t, y$ completes our proof. $\qquad\square$

## 3.2 TRACTABLE BOUND

It is worth noticing that bound (1) contains some unknown quantities, e.g., $u$ and $a_i$'s, which cannot be computed exactly. Meanwhile, the bound in Corollary 1 uses an unknown (optimal) partition. Those are their main limitations. The following bound overcomes such limitations.

**Theorem 3.2.** *Given the notations and assumption in Theorem 3.1, for any constants $\gamma \geq 1, \delta > 0$ and $\alpha \in [0, \frac{\gamma n(K + \gamma n)}{K(4n - 3)}]$, we have the following with probability at least $1 - \gamma^{-\alpha} - \delta$:*

$$F(P, \boldsymbol{h}) \leq F(\boldsymbol{S}, \boldsymbol{h}) + C\sqrt{\hat{u} \alpha \ln \gamma} + g_2(\delta/2) \tag{2}$$

*where $\hat{u} = \frac{\gamma}{2n} + \frac{\gamma^2}{2} \sum_{i=1}^{K} \left(\frac{n_i}{n}\right)^2 + \gamma^2 \sqrt{\frac{2}{n} \ln \frac{2K}{\delta}}$, $g_2(\delta) = \frac{C(1 + \sqrt{2})\sqrt{\ln(2K/\delta)}}{n} \sum_{i \in \boldsymbol{T}} \sqrt{n_i} + \frac{4C|\boldsymbol{T}| \ln(2K/\delta)}{n}$.*

This result follows directly from Theorem 3.1, where $g_2(\delta)$ serves as a simplified (and generally looser) surrogate for the earlier term $g(\Gamma, \boldsymbol{h}, \delta_2)$. The original bound in Theorem 3.1 is not directly computable because the quantities $u$ and $g$ depend on intractable distributional terms. In contrast, $\hat{u}$ and $g_2$ are computable approximations of $u$ and $g$, respectively. It is important to note that while $\hat{u}$ preserves the essential role of $u$ in capturing distributional complexity, the simplified $g_2$ no longer retains the fine-grained local structure encoded in $g$. Therefore, Bound (2 serves as providing certificate for a model, and may not be ideal for comparing models.

A key advantage of this computable formulation is that *the bound can be evaluated using only the training set*. Once a choice of $K$ and a partition $\Gamma$ is fixed, we can compute the counts $n_i$, identify the set $\boldsymbol{T}$, and directly evaluate Bound (2). This ease of computation makes the bound particularly practical and appealing for large-scale applications.

**A theoretical comparison with closely related bounds:** Although many model-dependent bounds (Kawaguchi et al., 2022; Than et al., 2025; Biggs & Guedj, 2022; Viallard et al., 2024; Lotfi et al., 2024a;b) have been proposed, our bound (2) has various advantages:

- *Mild assumption:* Our bound does not require stringent assumptions as in prior ones. Some prior bounds require stability (Li et al., 2024; Lei & Ying, 2020) or robustness (Xu & Mannor, 2012; Kawaguchi et al., 2022; Sokolić et al., 2017) of the learning algorithm. Those assumptions are often violated in practice, e.g. for the appearance of adversarial attacks (Zhou et al., 2022). Some theories (Lotfi et al., 2024a;b) assume that the hypothesis class is finite, which is restrictive. In contrast, our bound requires only i.i.d. assumption which also appears in most prior bounds.
- *Easy evaluation:* An evaluation of our bound (2) will be simple and does not require any modification to the model $\boldsymbol{h}$ of interest. This is a crucial advantage. Many prior theories require intermediate steps to change the model of interest into a suitable form. For example, state-of-the-art methods to compress NNs are required for (Zhou et al., 2019; Lotfi et al., 2022; Nadjahi et al., 2024); quantization for a model is required for (Lotfi et al., 2024a;b); finetuning (e.g. SubLoRA) is required for (Lotfi et al., 2024a;b). Those facts suggests that

evaluations for prior bounds are often expensive. Besides, many prior model-dependent bounds (Xu & Mannor, 2012; Kawaguchi et al., 2022; Than et al., 2025) cannot be exactly computed from a training set only.

- *No change to the model:* Most prior non-vacuous bounds (Zhou et al., 2019; Dziugaite & Roy, 2017; Lotfi et al., 2024a;b) require extensively compressing (or quantizing) model $h$ of interest and then retraining/finetuning the compressed version. Sometimes the compression step is too restrictive and produces low-quality models (Lotfi et al., 2024a). Therefore, a modification will change model $h$ and hence ***those bounds do not directly provide guarantees for the generalization ability of $h$***. In contrast, our bound (2) does not require any change to model $h$, and hence directly provides a guarantee for $h$.

*Remark* 2. There is a nonlinear relationship between $K$ and the uncertainty term $\text{Unc}(\Gamma) = C\sqrt{\hat{u}\alpha \ln \gamma} + g_2(\delta/2)$ in our bound. A partition with a larger $K$ can make the sum $\sum_{i=1}^{K} (n_i/n)^2$ smaller, as the samples can be spread into more areas. However a larger $K$ can make $g_2(\delta)$ larger. Therefore, we should not choose too large $K$. On the other hand, a small $K$ can make the sum $\sum_{i=1}^{K} (n_i/n)^2$ large, since more samples can appear in each area $\mathcal{Z}_i$ and enlarge $n_i/n$. Therefore, we should not choose too small $K$. Furthermore, we need to choose constant $\alpha$ carefully, since there is a trade-off in the bound and the certainty $1 - \gamma^{-\alpha} - \delta$. A smaller $\alpha$ can make the bound smaller, but could enlarge $\gamma^{-\alpha}$ and hence reduce the certainty of the bound.

## 4 EMPIRICAL EVALUATION

In this section, we present an extensive empirical evaluation of our bounds. We first investigate the strength of the guarantees they provide for the test error of trained models without any modification. We then examine how well our bounds correlate with model quality and how they are affected by key factors. More results appear in Appendix D.

### 4.1 MAIN RESULTS: GUARANTEES FOR LARGE-SCALE PRETRAINED MODELS

**Models:** We use 32 modern NN models[2] which were pretrained by Pytorch on the ImageNet dataset with 1,281,167 images. All models are multiclass classifiers. We use the ImageNet training set exclusively to compute the bound (2).

**Baselines:** While many model-dependent bounds exist, we exclude them from direct comparison for the following reasons: (1) several bounds (Kawaguchi et al., 2022; Than et al., 2025; von Luxburg & Bousquet, 2004; Hou et al., 2023) cannot be computed exactly from the training set alone; (2) all norm-based bounds (Bartlett et al., 2017; Arora et al., 2018; Golowich et al., 2020; Graf et al., 2022; Galanti et al., 2023b) are vacuous even for relatively small networks; and (3) certain PAC-Bayes bounds (Biggs & Guedj, 2022) apply only to shallow or specialized architectures, while others (Zhou et al., 2019; Dziugaite & Roy, 2017; Lotfi et al., 2022) estimate $\mathbb{E}_{\hat{h}}[F(P, \hat{h})]$, the expected test error of a stochastic model. Those bounds and the ones in (Lotfi et al., 2024a;b) require substantial modifications to the original network. Such requirements render them incompatible with our evaluation setting.

**Experimental settings:** We fix $\delta = 0.01, \alpha = 100, \gamma = 0.04^{-1/\alpha}$. This choice means that our bound is correct with probability at least 95%. The partition $\Gamma$ is chosen with $K = 200$ small areas of the input space, by clustering the training images into 200 areas, whose centroids are initialized randomly. The upper bound (2) for each model was computed with 5 random seeds. We use the 0-1 loss function, meaning that our bound directly estimates the true classification error.

**Results:** The overall results are reported in Table 2. One can observe that our bound for all models are all non-vacuous even for the non-optimized choices of some parameters. Our estimate is often 2-3 times higher than the oracle test error of each model. When choosing the best parameter for each model by grid search, we can obtain much better bounds about the test errors. Note that non-vacuousness of our bound holds true for a large class of deep NN families, some of which have more than 630M parameters. To the best of our knowledge, bound (2) is the first theoretical bound which is non-vacuous at such a large scale, without requiring any modification to the trained models.

---

[2]https://pytorch.org/vision/stable/models.html

Table 2: Upper bounds on the true error (in %) of 32 deep NNs which were pretrained on ImageNet dataset. The second column presents the model size, the third column contains the test accuracy at Top 1, as reported by Pytorch. "Mild" reports the bound for the choice of $\{\delta = 0.01, K = 200, \alpha = 100, \gamma = 0.04^{-1/\alpha}\}$, while "Optimized" reports the bound with parameter optimization by grid search. The grid search is done for $K \in \{100, 200, 300, 400, 500, 1000, 5000, 10000\}, \alpha \in \{10, 20, ..., 100\}, \delta = 0.01$ and $\gamma = 0.04^{-1/\alpha}$. The last two columns report our estimates about the true error, with a certainty at least 95%.

| Model | #Params (M) | Training error | Acc@1 | Test error | Error bound (2) | |
|---|---|---|---|---|---|---|
| | | | | | Mild | Optimized |
| ResNet18 V1 | 11.7 | 21.245 | 69.758 | 30.242 | 57.896 $_{\pm 4.189}$ | 54.262 |
| ResNet34 V1 | 21.8 | 15.669 | 73.314 | 26.686 | 52.320 $_{\pm 4.189}$ | 48.686 |
| ResNet50 V1 | 25.6 | 13.121 | 76.130 | 23.870 | 49.772 $_{\pm 4.189}$ | 46.138 |
| ResNet101 V1 | 44.5 | 10.502 | 77.374 | 22.626 | 47.153 $_{\pm 4.189}$ | 43.519 |
| ResNet152 V1 | 60.2 | 10.133 | 78.312 | 21.688 | 46.784 $_{\pm 4.189}$ | 43.150 |
| ResNet50 V2 | 25.6 | 8.936 | 80.858 | 19.142 | 45.587 $_{\pm 4.189}$ | 41.953 |
| ResNet101 V2 | 44.5 | 6.008 | 81.886 | 18.114 | 42.659 $_{\pm 4.189}$ | 39.025 |
| ResNet152 V2 | 60.2 | 5.178 | 82.284 | 17.716 | 41.829 $_{\pm 4.189}$ | 38.195 |
| SwinTransformer B | 87.8 | 6.464 | 83.582 | 16.418 | 43.115 $_{\pm 4.189}$ | 39.481 |
| SwinTransformer B V2 | 87.9 | 6.392 | 84.112 | 15.888 | 43.043 $_{\pm 4.189}$ | 39.409 |
| SwinTransformer T | 28.3 | 9.992 | 81.474 | 18.526 | 46.643 $_{\pm 4.189}$ | 43.009 |
| SwinTransformer T V2 | 28.4 | 8.724 | 82.072 | 17.928 | 45.375 $_{\pm 4.189}$ | 41.741 |
| VGG13 | 133.0 | 18.456 | 69.928 | 30.072 | 55.107 $_{\pm 4.189}$ | 51.473 |
| VGG13 BN | 133.1 | 19.223 | 71.586 | 28.414 | 55.874 $_{\pm 4.189}$ | 52.240 |
| VGG19 | 143.7 | 16.121 | 72.376 | 27.624 | 52.772 $_{\pm 4.189}$ | 49.138 |
| VGG19 BN | 143.7 | 15.941 | 74.218 | 25.782 | 52.592 $_{\pm 4.189}$ | 48.958 |
| DenseNet121 | 8.0 | 15.631 | 74.434 | 25.566 | 52.282 $_{\pm 4.189}$ | 48.648 |
| DenseNet161 | 28.7 | 10.48 | 77.138 | 22.862 | 47.131 $_{\pm 4.189}$ | 43.497 |
| DenseNet169 | 14.1 | 12.395 | 75.600 | 24.400 | 49.046 $_{\pm 4.189}$ | 45.412 |
| DenseNet201 | 20.0 | 9.806 | 76.896 | 23.104 | 46.457 $_{\pm 4.189}$ | 42.823 |
| ConvNext Base | 88.6 | 5.209 | 84.062 | 15.938 | 41.860 $_{\pm 4.189}$ | 38.226 |
| ConvNext Large | 197.8 | 3.846 | 84.414 | 15.586 | 40.497 $_{\pm 4.189}$ | 36.863 |
| RegNet Y 128GF linear | 644.8 | 9.032 | 86.068 | 13.932 | 45.683 $_{\pm 4.189}$ | 42.049 |
| RegNet Y 32GF linear | 145.0 | 10.558 | 84.622 | 15.378 | 47.209 $_{\pm 4.189}$ | 43.575 |
| RegNet Y 32GF V2 | 145.0 | 3.761 | 81.982 | 18.018 | 40.412 $_{\pm 4.189}$ | 36.778 |
| RegNet Y 32GF e2e | 145.0 | 7.127 | 86.838 | 13.162 | 43.778 $_{\pm 4.189}$ | 40.144 |
| RegNet Y 128GF e2e | 644.8 | 5.565 | 88.228 | 11.772 | 42.216 $_{\pm 4.189}$ | 38.582 |
| VIT H 14 linear | 632.0 | 9.951 | 85.708 | 14.292 | 46.602 $_{\pm 4.189}$ | 42.968 |
| VIT B 16 linear | 86.6 | 14.969 | 81.886 | 18.114 | 51.620 $_{\pm 4.189}$ | 47.986 |
| VIT L 16 linear | 304.3 | 11.003 | 85.146 | 14.854 | 47.654 $_{\pm 4.189}$ | 44.020 |
| VIT B 16 V1 | 86.6 | 5.916 | 81.072 | 18.928 | 42.567 $_{\pm 4.189}$ | 38.933 |
| VIT L 16 V1 | 304.3 | 3.465 | 79.662 | 20.338 | 40.116 $_{\pm 4.189}$ | 36.482 |

## 4.2 Ablation study

EFFECT OF PARAMETERS: Note that our bound depends on the choice of some parameters. Figure 1 reports the changes of $\sum_{i=1}^{K} \left(\frac{n_i}{n}\right)^2$ as the partition $\Gamma$ changes. We can see that this quantity tends to decrease as we divide the input space into more small areas. Meanwhile, Figure 2 reports the uncertainty term, as either $\alpha$ or $K$ changes. Observe that a larger $K$ can increase the uncertainty fast, while an increase in $\alpha$ can gradually decrease the uncertainty. Those figures enable an easy choice for the parameters in our bound.

IMPACT OF THE DATA AND PARTITIONING ALIGNMENT

Our bounds can be tighter if we can choose a good alignment between the partition and data distribution. However, analyzing the effect of data geometry and partitioning strategies is indeed challenging, particularly in high-dimensional settings with unknown data distributions. To address this, we designed two controlled ablations using a synthetic mixture model, described in Appendix D.1.

**Exploring different partitioning strategies:** We considered three types of partitions $\Gamma$:
**T1:** A uniform grid partition that divides the data space into equally sized regions. However, this strategy may not align with the actual data distribution, potentially resulting in regions with highly

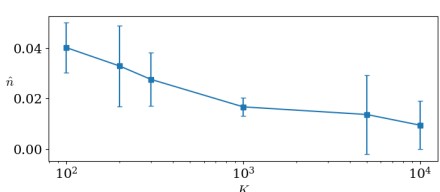 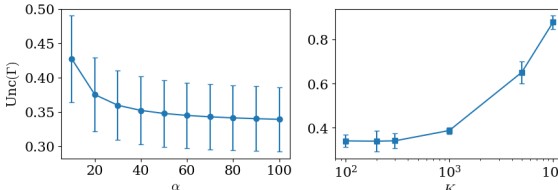

Figure 1: The dynamic of $\hat{n} = \sum_{i=1}^{K} \left( \frac{n_i}{n} \right)^2$ as $K$ changes.

Figure 2: The uncertainty $\text{Unc}(\Gamma) = C\sqrt{\hat{u}\alpha \ln \gamma} + g_2(\delta/2)$ as (right) $K$ changes and (left) $\alpha$ changes, for fixed $K = 200, \gamma = 0.04^{-1/\alpha}, \delta = 0.01$.

Table 3: Uncertainty term $Unc(\Gamma)$ under (a) *different partitioning strategies* (for fixed variance $\nu = 1$) and (b) *data geometries* (for fixed partition **T3**), as $\alpha$ changes. Smaller is better.

|  | (a) | | | |  |  | (b) | | | |
|---|---|---|---|---|---|---|---|---|---|---|
| $\alpha$ | 4 | 6 | 8 | 10 | | $\alpha$ | 4 | 6 | 8 | 10 |
| **T1** | 1.0045 | 0.8106 | 0.7315 | 0.6889 | | $\nu = 10^0$ | 0.8450 | 0.7147 | 0.6615 | 0.6328 |
| **T2** | 0.8723 | 0.7302 | 0.6722 | 0.6409 | | $\nu = 10^2$ | 0.8458 | 0.7151 | 0.6618 | 0.6331 |
| **T3** | 0.8450 | 0.7147 | 0.6615 | 0.6328 | | $\nu = 10^4$ | 1.0142 | 0.8393 | 0.7680 | 0.7295 |

imbalanced probability measures.

**T2:** A partition formed by uniformly generating $K$ centroids to define the regions. Like T1, this method may not capture the underlying structure of the data.

**T3:** A partition where the centroids of the mixture components are fixed as region centers. This approach tends to yield more balanced regions, where $p_i \approx p_j$ for all $i, j$, for small variances.

To evaluate the quality of these partitions, we generated $100000$ i.i.d. samples from the MM (with variance $\nu = 1$) and computed $Unc(\Gamma)$ for varying $\alpha$, with $K = 100$, $\gamma = 0.04^{-1/\alpha}$ and $\delta = 0.01$. The results are reported in Table 3(a). Among the three strategies, **T1** resulted in the highest uncertainty, while **T3** consistently produced the lowest. These findings suggest that partitions leading to balanced local measures (such as **T3**) are more favorable, while those poorly aligned with the data distribution (**T1**, **T2**) lead to higher uncertainty. This empirical evidence supports our theoretical discussion on the importance of selecting meaningful partitions.

**Exploring data geometries:** We further examine how the geometry of the data distribution influences the uncertainty term. To this end, we consider the same mixture model with varying variances $\nu \in \{10^0, 10^2, 10^4\}$ while fixing **T3** as the partition. Note that increasing $\nu$ to $10^4$ significantly alters the geometry of the mixture model compared to the case $\nu = 1$. The corresponding uncertainty values are reported in Table 3(b). These results demonstrate that $Unc(\Gamma)$ can vary considerably depending on the geometry induced by the data distribution. When the partition $\Gamma$ does not align well with the data, the resulting local regions may have highly imbalanced probability measures. In such cases, the uncertainty can be large.

## 5 CONCLUSION AND DISCUSSION

Understanding and certifying the behavior of modern deep networks remains a foundational challenge for reliable machine learning. This work introduces a new class of *data-dependent generalization bounds* that apply directly to trained models, without compression, architectural modification, or retraining. Among them, the exactly computable bound stands out: it is non-vacuous across all evaluated deep networks, including ImageNet-scale models, demonstrating that meaningful, practical guarantees are achievable even for large, unaltered networks. This closes part of a longstanding gap in learning theory by showing that high-capacity models can admit informative certificates when analysis is aligned with the *geometry of the data distribution* rather than the parameter space.

A central insight of our framework is that generalization can be decomposed into two interpretable components:

- a *distributional complexity term*, capturing how concentrated or diffuse the data distribution is across the partition; and

- *local model-behavior terms*, capturing how the trained network behaves in specific regions of the input space.

This joint dependence does not appear in classical bounds. It reveals where the generalization gap arises and why, highlighting the specific regions and local behaviors responsible. Empirically, we observe in Appendix D that components of Bound (1), such as the local-loss–weighted concentration term, are highly predictive of the true test error. Moreover, the bound tightens significantly when the partition aligns with the intrinsic data geometry, underscoring that **data-dependent local regularity** is a key driver of modern generalization. This interpretability is complementary to classical theories and offers a new axis along which to understand large models.

At the same time, our framework has some limitations that point to fertile directions for future research. The bounds can be loose when partitions are poorly aligned with the data distribution or when sample sizes are small, since multinomial counts may be far from their expectations. Likewise, when the model underfits the data, the empirical loss dominates the bound. These failure modes clarify that the approach naturally excels in high-data, high-capacity regimes (e.g. ImageNet) while classical bounds may remain more effective for small datasets or highly structured models.

Looking forward, several extensions would strengthen the theory. One direction is to integrate additional model properties to refine the uncertainty terms of Theorem 3.1, potentially reducing the gap in regimes where data alone are insufficient. Another avenue is to extend the analysis to non-i.i.d. samples, which would require advances in concentration inequalities beyond the i.i.d. setting. Finally, because the theory is agnostic to architecture, applying it to other domains such as regression, language inference, or text-to-image generation may reveal new types of geometry-inspired theories.

In summary, while not universal, the proposed framework provides a novel informative perspective on why large modern models generalize and offers concrete, interpretable quantities that both diagnose and bound generalization behavior. We hope this work serves as a foundation for future research combining data-centric and classical theories in a unified understanding of generalization.

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

## A    PROOFS FOR MAIN RESULTS

*Proof of Theorem 3.1.* We first observe that

$$F(P, \boldsymbol{h}) - F(\boldsymbol{S}, \boldsymbol{h}) = F(P, \boldsymbol{h}) - \sum_{i=1}^{K} \frac{n_i}{n} a_i(\boldsymbol{h}) + \sum_{i=1}^{K} \frac{n_i}{n} a_i(\boldsymbol{h}) - F(\boldsymbol{S}, \boldsymbol{h}) \tag{3}$$

Next, we consider $F(P, \boldsymbol{h}) - \sum_{i=1}^{K} \frac{n_i}{n} a_i(\boldsymbol{h}) = \sum_{i=1}^{K} p_i a_i(\boldsymbol{h}) - \sum_{i=1}^{K} \frac{n_i}{n} a_i(\boldsymbol{h}) = \sum_{i=1}^{K} a_i(\boldsymbol{h}) \left[ p_i - \frac{n_i}{n} \right]$. Note that $(n_1, ..., n_K)$ is a multinomial random variable with parameters $n$ and $(p_1, ..., p_K)$. Therefore, according to Lemma 7 in (Kawaguchi et al., 2022), we have $\Pr\left( \sum_{i=1}^{K} a_i(\boldsymbol{h}) \left[ p_i - \frac{n_i}{n} \right] > g(\Gamma, \boldsymbol{h}, \delta_2) \right) < \delta_2$. This implies

$$\Pr\left( F(P, \boldsymbol{h}) - \sum_{i=1}^{K} \frac{n_i}{n} a_i(\boldsymbol{h}) > g(\Gamma, \boldsymbol{h}, \delta_2) \right) < \delta_2 \tag{4}$$

On the other hand, Theorem A.1 below shows that

$$\Pr\left( \sum_{i \in \boldsymbol{T}_S} \frac{n_i}{n} a_i(\boldsymbol{h}) - F(\boldsymbol{S}, \boldsymbol{h}) \geq C\sqrt{\frac{u}{2n^2} \ln \frac{1}{\delta_1}} \right) \leq \delta_1 \tag{5}$$

Combining this with (4) and the union bound, we have

$$\Pr\left( F(P, \boldsymbol{h}) > F(\boldsymbol{S}, \boldsymbol{h}) + C\sqrt{\frac{u}{2n^2} \ln \frac{1}{\delta_1}} + g(\Gamma, \boldsymbol{h}, \delta_2) \right) < \delta_1 + \delta_2 \tag{6}$$

completing the proof.  $\square$

*Proof of Corollary 1.* A simple consequence of using quantiles for continuous distributions (Hallin et al., 2021; Figalli, 2018) suggests that there exists a partition $\Gamma^*(\mathcal{Z}) := \bigcup_{i=1}^{K} \mathcal{Z}_i^*$ so that $P(\mathcal{Z}_i^*) = \frac{1}{K}, \forall i \in [K]$. The result of this corollary can be derived by applying Theorem 3.1 for parirition $\Gamma^*$, where $p_i = 1/K$ for all $i$.  $\square$

*Proof of Theorem 3.2.* Theorem 3.1 shows that

$$\Pr\left( F(P, \boldsymbol{h}) > F(\boldsymbol{S}, \boldsymbol{h}) + C\sqrt{\frac{u}{2n^2} \ln \frac{1}{\delta_1}} + g(\Gamma, \boldsymbol{h}, \delta/2) \right) < \delta_1 + \delta/2 \tag{7}$$

where $u$ and $\delta_1$ depend on the sum $\sum_{i=1}^{K} p_i^2$. We next bound this quantity using $\boldsymbol{S}$.

Since $p_i \geq 0$ and $\sum_{i=1}^{K} p_i = 1$, we can use the Lagrange multiplier method to show that $\sum_{i=1}^{K} p_i^2$ is minimized at $1/K$. Hence $u = \sum_{i=1}^{K} \gamma n p_i (1 + \gamma n p_i) = \gamma n + \gamma^2 n^2 \sum_{i=1}^{K} p_i^2 \geq \gamma n + \gamma^2 n^2 / K$. This suggests that $\exp(-\frac{u \ln \gamma}{4n-3}) \leq \exp(-\frac{(\gamma n + \gamma^2 n^2 / K) \ln \gamma}{4n-3}) \leq \exp(-\frac{\gamma n (K + \gamma n) \ln \gamma}{K(4n-3)}) \leq \gamma^{-\alpha}$. Choosing $\delta_1 = \gamma^{-\alpha}$ and plugging it into (7) lead to

$$\Pr\left( F(P, \boldsymbol{h}) > F(\boldsymbol{S}, \boldsymbol{h}) + C\sqrt{\frac{u}{2n^2} \alpha \ln \gamma} + g(\Gamma, \boldsymbol{h}, \delta/2) \right) < \delta/2 + \gamma^{-\alpha} \tag{8}$$

It is easy to see that $g(\Gamma, \boldsymbol{h}, \delta/2) \leq g_2(\delta/2)$, since $a_o(\boldsymbol{h}) \leq C$ and $a_i(\boldsymbol{h}) \leq C$ for any $i$. Therefore

$$\Pr\left( F(P, \boldsymbol{h}) > F(\boldsymbol{S}, \boldsymbol{h}) + C\sqrt{\frac{u}{2n^2} \alpha \ln \gamma} + g_2(\delta/2) \right) < \delta/2 + \gamma^{-\alpha} \tag{9}$$

Next we consider $\frac{u}{2n^2} = \frac{\gamma}{2n} + \frac{\gamma^2}{2} \sum_{i=1}^{K} p_i^2$. Since $\boldsymbol{S}$ contains $n$ i.i.d. samples, $(n_1, ..., n_K)$ is a multinomial random variable with parameters $n$ and $(p_1, ..., p_K)$. Lemma B.7 shows

$$\Pr\left( \sum_{i=1}^{K} p_i^2 > \sum_{i=1}^{K} \left( \frac{n_i}{n} \right)^2 + 2\sqrt{\frac{2}{n} \ln \frac{2K}{\delta}} \right) < \delta/2$$

Therefore $\Pr\left(\frac{u}{2n^2} > \frac{\gamma}{2n} + \frac{\gamma^2}{2}\sum_{i=1}^{K}\left(\frac{n_i}{n}\right)^2 + \gamma^2\sqrt{\frac{2}{n}\ln\frac{2K}{\delta}}\right) < \delta/2$. This also suggests that

$$\Pr\left(C\sqrt{\frac{u}{2n^2}\alpha\ln\gamma} > C\sqrt{\hat{u}\alpha\ln\gamma}\right) < \delta/2 \tag{10}$$

Combining this with (9) and the union bound will complete the proof. □

## A.1 APPROXIMATING THE INTRACTABLE PART BY A DATA SET

**Theorem A.1.** *Given the notations in Theorem 3.1,*

$$\Pr\left(\sum_{i\in\boldsymbol{T}_S}\frac{n_i}{n}a_i(\boldsymbol{h}) \geq \sum_{i\in\boldsymbol{T}_S}\frac{n_i}{n}F(\boldsymbol{S}_i,\boldsymbol{h}) + C\sqrt{\frac{u}{2n^2}\ln\frac{1}{\delta_1}}\right) \leq \delta_1 \tag{11}$$

*Proof.* Denote $\boldsymbol{n} = \{n_1, ..., n_K\}$ and for each $j \in [K]$:

$$B_j = \sum_{i=1}^{j} n_i a_i(\boldsymbol{h}) - \sum_{i=1}^{j} n_i F(\boldsymbol{S}_i, \boldsymbol{h}) \tag{12}$$

$$X_j = n_j F(\boldsymbol{S}_j, \boldsymbol{h}) \tag{13}$$

$$\boldsymbol{S}_{\leq j} = \bigcup_{i\leq j} \boldsymbol{S}_i \tag{14}$$

Denote $y = \frac{4t}{uC^2}$ for any $t \in \left[0, uC\sqrt{\frac{\ln\gamma}{8n-6}}\right]$. The proof for (11) contains three main steps.

**Step 1:** We first observe that

$$\Pr\left(B_K \geq t\right) \leq e^{-yt}\mathbb{E}_{\boldsymbol{S}}\left[e^{yB_K}\right] \qquad\qquad \text{(Chernoff bounds)} \tag{15}$$

$$\leq e^{-yt}\mathbb{E}_{\boldsymbol{h},\boldsymbol{n}}\left[\mathbb{E}_{\boldsymbol{S}}\left[e^{yB_K}|\boldsymbol{h},\boldsymbol{n}\right]\right] \qquad \text{(Law of total expectation)} \tag{16}$$

**Step 2 - estimating $\mathbb{E}_{\boldsymbol{S}}\left[e^{yB_K}|\boldsymbol{h},\boldsymbol{n}\right]$:** We observe the following for each $j \in \boldsymbol{T}_S$,

$$\mathbb{E}_{X_j}[X_j|\boldsymbol{h},\boldsymbol{n}] = \mathbb{E}_{\boldsymbol{S}_j}[n_j F(\boldsymbol{S}_j,\boldsymbol{h})|\boldsymbol{h},\boldsymbol{n}] \tag{17}$$

$$= \mathbb{E}_{\boldsymbol{S}_j}\left[\sum_{i=1}^{n_j}\ell(\boldsymbol{h},\boldsymbol{z}_{ji})|\boldsymbol{h},\boldsymbol{n}\right] \qquad \text{(where } \boldsymbol{S}_j = \{\boldsymbol{z}_{ji}\}_{i=1}^{n_j}) \tag{18}$$

$$= \sum_{i=1}^{n_j}\mathbb{E}_{\boldsymbol{z}_{ji}\in\mathcal{Z}_j}\left[\ell(\boldsymbol{h},\boldsymbol{z}_{ji})|\boldsymbol{h},\boldsymbol{n}\right] \qquad (\boldsymbol{S}_j \text{ contains i.i.d. samples in } \mathcal{Z}_j) \tag{19}$$

$$= \sum_{i=1}^{n_j} a_j(\boldsymbol{h}) = n_j a_j(\boldsymbol{h}) \tag{20}$$

Therefore $B_j = B_{j-1} + \mathbb{E}_{X_j}[X_j|\boldsymbol{h},\boldsymbol{n}] - X_j$ for all $j \in \boldsymbol{T}_S$. Note that $B_i = B_{i-1}$ (due to $n_i = b_i = X_i = 0$) for all $i \notin \boldsymbol{T}_S$. Hence, for $i \notin \boldsymbol{T}_S$, we will use $\mathbb{E}_{X_i}[X_i|\boldsymbol{h},\boldsymbol{n}] - X_i$ instead of 0 in the below analysis for simplicity of presentation.

We can rewrite

$$\mathbb{E}_{\boldsymbol{S}}\left[e^{yB_K}|\boldsymbol{h},\boldsymbol{n}\right] = \mathbb{E}_{\boldsymbol{S}}\left[e^{y(B_{K-1}+\mathbb{E}_{X_K}[X_K|\boldsymbol{h},\boldsymbol{n}]-X_K)}|\boldsymbol{h},\boldsymbol{n}\right] \tag{21}$$

$$= \mathbb{E}_{\boldsymbol{S}_{\leq K}}\left[e^{y(B_{K-1}+\mathbb{E}_{X_K}[X_K|\boldsymbol{h},\boldsymbol{n}]-X_K)}|\boldsymbol{h},\boldsymbol{n}\right] \tag{22}$$

$$\leq \mathbb{E}_{\boldsymbol{S}_{\leq K-1}}\left[e^{yB_{K-1}}|\boldsymbol{h},\boldsymbol{n}\right]\mathbb{E}_{X_K}\left[e^{y(\mathbb{E}_{X_K}[X_K|\boldsymbol{h},\boldsymbol{n}]-X_K)}|\boldsymbol{h},\boldsymbol{n}\right] \tag{23}$$

where the last inequality comes from the fact that $X_K$ is conditionally independent with $\boldsymbol{S}_{\leq K-1}$, conditioned on $\{\boldsymbol{h},\boldsymbol{n}\}$.

It is easy to see that $0 \leq X_K \leq Cn_K$, due to $0 \leq F(\boldsymbol{S}_K, \boldsymbol{h}) \leq C$. Lemma B.1 implies $\mathbb{E}_{X_K}\left[e^{y(\mathbb{E}_{X_K}[X_K|\boldsymbol{h},\boldsymbol{n}]-X_K)}|\boldsymbol{h},\boldsymbol{n}\right] \leq \exp\left(\frac{y^2C^2n_K^2}{8}\right)$. Plugging this into (23), we obtain

$$\mathbb{E}_{\boldsymbol{S}}\left[e^{yB_K}|\boldsymbol{h},\boldsymbol{n}\right] \leq \mathbb{E}_{\boldsymbol{S}_{\leq K-1}}\left[e^{yB_{K-1}}|\boldsymbol{h},\boldsymbol{n}\right]\exp\left(\frac{y^2C^2n_K^2}{8}\right) \tag{24}$$

Using the same arguments for $X_{K-1}, ..., X_1$, we obtain the followings

$$\mathbb{E}_{\boldsymbol{S}}\left[e^{yB_K}|\boldsymbol{h},\boldsymbol{n}\right] \leq \mathbb{E}_{\boldsymbol{S}_{\leq K-2}}\left[e^{yB_{K-2}}|\boldsymbol{h},\boldsymbol{n}\right]\exp\left(\frac{y^2C^2n_K^2}{8} + \frac{y^2C^2n_{K-1}^2}{8}\right)$$

$$...$$

$$\leq \exp\left(\frac{y^2C^2}{8}\sum_{i=1}^{K}n_i^2\right) \tag{25}$$

**Step 3 - bounding** $\Pr\left(B_K \geq t\right)$**:** By combining (25) with (16), we obtain

$$\Pr\left(B_K \geq t\right) \leq e^{-yt}\mathbb{E}_{\boldsymbol{h},\boldsymbol{n}}\exp\left(\frac{y^2C^2}{8}\sum_{i=1}^{K}n_i^2\right) \tag{26}$$

$$= e^{-yt}\mathbb{E}_{\boldsymbol{n}}\exp\left(\frac{y^2C^2}{8}\sum_{i=1}^{K}n_i^2\right) \tag{27}$$

$$\leq e^{-yt}\mathbb{E}_{\boldsymbol{n}}\exp\left(\frac{y^2C^2}{8}\sum_{i=1}^{K-1}n_i^2\right)\mathbb{E}_{n_K}\exp\left(\frac{y^2C^2}{8}n_K^2\right) \tag{28}$$

$$(\text{Since } n_K \text{ is independent with } n_1, ..., n_{K-1})$$

When $\gamma p_K < 1$, due to $t \leq uC\sqrt{\frac{\ln\gamma}{8n-6}}$, observe that $\frac{y^2C^2}{8} = \frac{2t^2}{u^2C^2} \leq \frac{\ln\gamma}{4n-3} \leq \frac{\ln\gamma}{(1-\gamma p_K)(4n-3)}$. Note that $n_K$ is a binomial random variable with parameters $n$ and $p_K$. Combining those facts with Lemma B.6 implies $\mathbb{E}_{n_K}\exp\left(\frac{y^2C^2}{8}n_K^2\right) \leq \exp\left(\frac{y^2C^2}{8}\gamma np_K(1+\gamma np_K)\right)$. On the other hand, Lemma B.5 also implies $\mathbb{E}_{n_K}\exp\left(\frac{y^2C^2}{8}n_K^2\right) \leq \exp\left(\frac{y^2C^2}{8}\gamma np_K(1+\gamma np_K)\right)$ when $\gamma p_K \geq 1$. As a result, those facts and (28) lead to the following:

$$\Pr\left(B_K \geq t\right) \leq e^{-yt}\mathbb{E}_{\boldsymbol{n}}\exp\left(\frac{y^2C^2}{8}\sum_{i=1}^{K-1}n_i^2\right)\exp\left(\frac{y^2C^2}{8}\left((1+\gamma np_K)\gamma np_K\right)\right) \tag{29}$$

Using the same arguments for the remaining variables $n_{K-1}, ..., n_1$, we obtain

$$\Pr\left(B_K \geq t\right) \leq \exp\left(-yt + \frac{y^2C^2}{8}\sum_{i=1}^{K}(1+\gamma np_i)\gamma np_i\right) \tag{30}$$

$$= \exp\left(-yt + \frac{y^2C^2u}{8}\right) = \exp\left(\frac{-2t^2}{uC^2}\right) \tag{31}$$

As a result

$$\Pr\left(\sum_{i=1}^{K}n_ia_i(\boldsymbol{h}) \geq \sum_{i=1}^{K}n_iF(\boldsymbol{S}_i,\boldsymbol{h}) + t\right) \leq \exp\left(-\frac{2t^2}{uC^2}\right) \tag{32}$$

Since $n_j = 0$ for all $j \notin \boldsymbol{T}_S$, we have

$$\Pr\left(\sum_{i\in\boldsymbol{T}_S}n_ia_i(\boldsymbol{h}) \geq \sum_{i\in\boldsymbol{T}_S}n_iF(\boldsymbol{S}_i,\boldsymbol{h}) + t\right) \leq \exp\left(-\frac{2t^2}{uC^2}\right) \tag{33}$$

Multiplying both sides (of the probability term) with $1/n$ leads to

$$\Pr\left(\sum_{i\in\boldsymbol{T}_S}\frac{n_i}{n}a_i(\boldsymbol{h}) \geq \sum_{i\in\boldsymbol{T}_S}\frac{n_i}{n}F(\boldsymbol{S}_i,\boldsymbol{h}) + t/n\right) \leq \exp\left(-\frac{2t^2}{uC^2}\right)$$

Choosing $t = C\sqrt{\frac{u}{2}\ln\frac{1}{\delta_1}}$ results in (11), completing the proof. $\qquad\square$

# B    SUPPORTING THEOREMS AND LEMMAS

## B.1    HOEFFDING'S LEMMA

**Lemma B.1** (Hoeffding's lemma for conditionals)**.** *Let $X$ be any real-valued random variable that may depend on some random variables $\boldsymbol{Y}$. Assume that $a \leq X \leq b$ almost surely, for some constants $a, b$. Then, for all $\lambda \in \mathbb{R}$,*

$$\mathbb{E}_X \left[ e^{\lambda(\mathbb{E}_X[X|\boldsymbol{Y}]-X)} | \boldsymbol{Y} \right] \leq \exp\left( \frac{\lambda^2(b-a)^2}{8} \right) \tag{34}$$

*Proof.* Denote $c = \mathbb{E}_X[X|\boldsymbol{Y}] - b, d = \mathbb{E}_X[X|\boldsymbol{Y}] - a$ and hence $c \leq 0 \leq d$.

Since $\exp$ is a convex function, we have the following for all $\mathbb{E}_X[X|\boldsymbol{Y}] - X \in [c, d]$:

$$e^{\lambda(\mathbb{E}_X[X|\boldsymbol{Y}]-X)} \leq \frac{d - \mathbb{E}_X[X|\boldsymbol{Y}] + X}{d - c}e^{\lambda c} + \frac{\mathbb{E}_X[X|\boldsymbol{Y}] - X - c}{d - c}e^{\lambda d}$$

Therefore, by taking the conditional expectation over $X$ for both sides,

$$\begin{aligned}
\mathbb{E}_X \left[ e^{\lambda(\mathbb{E}_X[X|\boldsymbol{Y}]-X)} | \boldsymbol{Y} \right] &\leq \frac{d - \mathbb{E}_X[X|\boldsymbol{Y}] + \mathbb{E}_X[X|\boldsymbol{Y}]}{d - c}e^{\lambda c} + \frac{\mathbb{E}_X[X|\boldsymbol{Y}] - \mathbb{E}_X[X|\boldsymbol{Y}] - c}{d - c}e^{\lambda d} \\
&= \frac{d}{d - c}e^{\lambda c} - \frac{c}{d - c}e^{\lambda d} \tag{35} \\
&= e^{L(\lambda(d-c))} \tag{36}
\end{aligned}$$

where $L(h) = \frac{ch}{d-c} + \ln(1 + \frac{c - e^h c}{d - c})$. For this function, note that

$$L(0) = L'(0) = 0 \text{ and } L''(h) = -\frac{cde^h}{(d - ce^h)^2}$$

The AM-GM inequality suggests that $L''(h) \leq 1/4$ for all $h$. Combining this property with Taylor's theorem leads to the following, for some $\theta \in [0, 1]$,

$$L(h) = L(0) + hL'(0) + \frac{1}{2}h^2 L''(h\theta) \leq \frac{h^2}{8}$$

Combining this with (36) completes the proof.    □

## B.2    SMALL RANDOM VARIABLES

**Lemma B.2.** *Let $x_1, ..., x_n$ be independent random variables in $[0, 1]$ and satisfy $\mathbb{E}[x_i] \leq \nu, \forall i$ for some $\nu \in [0, 1]$. For any $c \geq 1$ satisfying $c\nu \geq 1$ and any $\lambda \geq 0$, we have $\mathbb{E} \exp\left(\lambda(x_1 + \cdots + x_n)^2\right) \leq \exp(\lambda cn\nu(1 + cn\nu))$.*

**Lemma B.3.** *Let $x_1, ..., x_n$ be independent random variables in $[0, 1]$ and satisfy $\mathbb{E}[x_i] \leq \nu, \forall i$ for some $\nu \in [0, 1]$. For any $c \geq 1$ satisfying $c\nu < 1$ and any $\lambda \in [0, \frac{\ln c}{(1-c\nu)(4n-3)}]$, we have $\mathbb{E} \exp\left(\lambda(x_1 + \cdots + x_n)^2\right) \leq \exp(\lambda cn\nu(1 + cn\nu))$.*

In order to prove those results, we need the following observations.

**Lemma B.4.** *Consider a random variable $X \in [0, 1]$ with mean $\mathbb{E}[X] \leq \nu$ for some constant $\nu \in [0, 1]$. For any $c \geq 1, \lambda \geq 0$:*

- *If $c\nu \geq 1$, then $\mathbb{E}e^{\lambda X} \leq e^{c\nu\lambda}$.*
- *If $c\nu < 1$, then $\mathbb{E}e^{\lambda X} \leq e^{c\nu\lambda}$ for all $\lambda \in [0, \frac{\ln c}{1-c\nu}]$.*

*Proof.* The Taylor series expansion of the function $e^{\lambda X}$ at any $X$ is $e^{\lambda X} = 1 + \sum_{p=1}^{\infty} \frac{(\lambda X)^p}{p!}$. Therefore

$$\mathbb{E}[e^{\lambda X}] = 1 + \sum_{p=1}^{\infty} \frac{\lambda^p}{p!} \mathbb{E}(X^p) \leq 1 + \mathbb{E}(X) \sum_{p=1}^{\infty} \frac{\lambda^p}{p!} \qquad \text{(due to } X^p \leq X, \forall p \geq 1) \quad (37)$$

$$\leq 1 + \nu \sum_{p=1}^{\infty} \frac{\lambda^p}{p!} = 1 + \nu(e^{\lambda} - 1) = 1 - \nu + \nu e^{\lambda} \quad (38)$$

Next we consider function $y(\lambda) = e^{c\nu\lambda} - 1 + \nu - \nu e^{\lambda}$. Its derivative is $y' = c\nu e^{c\nu\lambda} - \nu e^{\lambda} = \nu e^{\lambda}(ce^{(c\nu-1)\lambda} - 1)$.

For the case $c\nu \geq 1$, one can observe that $y' \geq 0$ for all $\lambda \geq 0$. This means $y$ is non-decreasing, and hence $y(\lambda) \geq y(0) = 0$. As a result, $e^{c\nu\lambda} \geq 1 - \nu + \nu e^{\lambda} \geq \mathbb{E}[e^{\lambda X}]$.

Consider the case $c\nu < 1$, it is easy to show that $y'(\lambda) \geq 0$ for all $\lambda \in [0, \frac{\ln c}{1-c\nu}]$. This means $y$ is non-decreasing in the interval $[0, \frac{\ln c}{1-c\nu}]$, and hence $y(\lambda) \geq y(0) = 0$ for all $\lambda \in [0, \frac{\ln c}{1-c\nu}]$. As a result, $e^{c\nu\lambda} \geq 1 - \nu + \nu e^{\lambda} \geq \mathbb{E}[e^{\lambda X}]$, completing the proof. □

**Corollary 2.** *Consider a random variable $X \in [0, 1]$ with mean $\mathbb{E}[X] \leq \nu$ for some constant $\nu \in [0, 1]$. For all constants $a, b \geq 0, c \geq 1$:*

- $\mathbb{E}e^{\lambda(aX^2+bX)} \leq e^{c(a+b)\nu\lambda}$, *for all $\lambda \geq 0$, if $c\nu \geq 1$.*
- $\mathbb{E}e^{\lambda(aX^2+bX)} \leq e^{c(a+b)\nu\lambda}$, *for all $\lambda \in [0, \frac{\ln c}{(1-c\nu)(a+b)}]$, if $c\nu < 1$.*

*Proof.* It is easy to observe that $\mathbb{E}e^{\lambda(aX^2)} \leq \mathbb{E}e^{\lambda(aX)}$ due to $X \in [0, 1]$. This suggests that $\mathbb{E}e^{\lambda(aX^2+bX)} \leq \mathbb{E}e^{\lambda(a+b)X}$. Applying Lemma B.4 will complete the proof. □

*Proof of Lemma B.2.* Denote $y_n = x_1 + \cdots + x_n$. Observe that $y_n = y_{n-1} + x_n$ and

$$\mathbb{E}_{y_n} e^{\lambda y_n^2} = \mathbb{E}_{y_n} e^{\lambda(y_{n-1}^2 + 2x_n y_{n-1} + x_n^2)} = \mathbb{E}_{y_{n-1}} \left[ e^{\lambda y_{n-1}^2} \mathbb{E}_{x_n} e^{\lambda(2x_n y_{n-1} + x_n^2)} \right] \quad (39)$$

Since $c\nu \geq 1$ and $x_n$ is independent with $y_{n-1}$, Corollary 2 implies $\mathbb{E}_{x_n} e^{\lambda(2x_n y_{n-1} + x_n^2)} \leq e^{c\nu\lambda(2y_{n-1}+1)}$. Plugging this into (39) leads to

$$\mathbb{E}_{y_n} e^{\lambda y_n^2} \leq \mathbb{E}_{y_{n-1}} \left[ e^{\lambda y_{n-1}^2} e^{c\nu\lambda(2y_{n-1}+1)} \right] = e^{c\nu\lambda} \mathbb{E}_{y_{n-1}} \left[ e^{\lambda(y_{n-1}^2 + 2c\nu y_{n-1})} \right] \quad (40)$$

Next we consider $\mathbb{E}_{y_{n-1}} \left[ e^{\lambda(y_{n-1}^2 + 2c\nu y_{n-1})} \right]$. Observe that $y_{n-1} = y_{n-2} + x_{n-1}$ and hence

$$\mathbb{E}_{y_{n-1}} \left[ e^{\lambda(y_{n-1}^2 + 2c\nu y_{n-1})} \right] = \mathbb{E}_{y_{n-1}} e^{\lambda(y_{n-2}^2 + 2x_{n-1}y_{n-2} + x_{n-1}^2 + 2c\nu x_{n-1} + 2c\nu y_{n-2})} \quad (41)$$

$$= \mathbb{E}_{y_{n-2}} \left[ e^{\lambda(y_{n-2}^2 + 2c\nu y_{n-2})} \mathbb{E}_{x_{n-1}} e^{\lambda(2x_{n-1}y_{n-2} + 2c\nu x_{n-1} + x_{n-1}^2)} \right] \quad (42)$$

Since $c\nu \geq 1$ and $x_{n-1}$ is independent with $y_{n-2}$, Corollary 2 implies $\mathbb{E}_{x_{n-1}} e^{\lambda(2x_{n-1}y_{n-2} + 2c\nu x_{n-1} + x_{n-1}^2)} \leq e^{c\nu\lambda(2y_{n-2}+2c\nu+1)}$. Plugging this into (42) leads to

$$\mathbb{E}_{y_{n-1}} \left[ e^{\lambda(y_{n-1}^2 + 2c\nu y_{n-1})} \right] \leq \mathbb{E}_{y_{n-2}} \left[ e^{\lambda(y_{n-2}^2 + 2c\nu y_{n-2})} e^{c\nu\lambda(2y_{n-2}+2c\nu+1)} \right] \quad (43)$$

$$= e^{c\nu\lambda(2c\nu+1)} \mathbb{E}_{y_{n-2}} \left[ e^{\lambda(y_{n-2}^2 + 4c\nu y_{n-2})} \right] \quad (44)$$

By using the same arguments, we can show that

$$\mathbb{E}_{y_{n-1}} \left[ e^{\lambda(y_{n-1}^2 + 2c\nu y_{n-1})} \right] \leq e^{c\nu\lambda(2c\nu+1)} e^{c\nu\lambda(4c\nu+1)} \mathbb{E}_{y_{n-3}} \left[ e^{\lambda(y_{n-3}^2 + 6c\nu y_{n-3})} \right] \quad (45)$$

$$= e^{2c\nu\lambda(3c\nu+1)} \mathbb{E}_{y_{n-3}} \left[ e^{\lambda(y_{n-3}^2 + 6c\nu y_{n-3})} \right] \quad (46)$$

$$...$$

$$\leq e^{c(n-2)\nu\lambda(c(n-1)\nu+1)} \mathbb{E}_{y_1} \left[ e^{\lambda(y_1^2 + 2c(n-1)\nu y_1)} \right] \quad (47)$$

Note that $\mathbb{E}_{y_1}\left[e^{\lambda(y_1^2+2c(n-1)\nu y_1)}\right] = \mathbb{E}_{x_1}\left[e^{\lambda(x_1^2+2c(n-1)\nu x_1)}\right] \leq e^{c\nu\lambda(1+2c(n-1)\nu)}$, according to Corollary 2. Combining this with (47), we obtain

$$\mathbb{E}_{y_{n-1}}\left[e^{\lambda(y_{n-1}^2+2c\nu y_{n-1})}\right] \leq e^{c(n-2)\nu\lambda(c(n-1)\nu+1)}e^{c\nu\lambda(1+2c(n-1)\nu)} = e^{c\nu\lambda(1+cn\nu)(n-1)} \tag{48}$$

By plugging this into (40), we obtain

$$\mathbb{E}_{y_n}e^{\lambda y_n^2} \leq e^{c\nu\lambda}e^{c\nu\lambda(1+cn\nu)(n-1)} = e^{c\nu\lambda((1+cn\nu)n-cn\nu)} \tag{49}$$
$$\leq e^{cn\nu(1+cn\nu)\lambda} \tag{50}$$

completing the proof. $\qquad\square$

*Proof of Lemma B.3.* Denote $y_n = x_1 + \cdots + x_n$ and observe that

$$\mathbb{E}_{y_n}e^{\lambda y_n^2} = \mathbb{E}_{y_n}e^{\lambda(y_{n-1}^2+2x_n y_{n-1}+x_n^2)} = \mathbb{E}_{y_{n-1}}\left[e^{\lambda y_{n-1}^2}\mathbb{E}_{x_n}e^{\lambda(2x_n y_{n-1}+x_n^2)}\right] \tag{51}$$

Note that $y_{n-1} = x_1 + \cdots + x_{n-1} \leq n-1$ and $\lambda(2y_{n-1}+1) \leq \lambda(2n-1) \leq \lambda(4n-3) \leq \frac{\ln c}{1-c\nu}$. Since $x_n$ is independent with $y_{n-1}$, Corollary 2 implies $\mathbb{E}_{x_n}e^{\lambda(2x_n y_{n-1}+x_n^2)} \leq e^{c\nu\lambda(2y_{n-1}+1)}$. Plugging this into (51) leads to

$$\mathbb{E}_{y_n}e^{\lambda y_n^2} \leq \mathbb{E}_{y_{n-1}}\left[e^{\lambda y_{n-1}^2}e^{c\nu\lambda(2y_{n-1}+1)}\right] = e^{c\nu\lambda}\mathbb{E}_{y_{n-1}}\left[e^{\lambda(y_{n-1}^2+2c\nu y_{n-1})}\right] \tag{52}$$

Next we consider $\mathbb{E}_{y_{n-1}}\left[e^{\lambda(y_{n-1}^2+2c\nu y_{n-1})}\right]$. Observe that

$$\mathbb{E}_{y_{n-1}}\left[e^{\lambda(y_{n-1}^2+2c\nu y_{n-1})}\right] = \mathbb{E}_{y_{n-1}}e^{\lambda(y_{n-2}^2+2x_{n-1}y_{n-2}+x_{n-1}^2+2c\nu x_{n-1}+2c\nu y_{n-2})} \tag{53}$$
$$= \mathbb{E}_{y_{n-2}}\left[e^{\lambda(y_{n-2}^2+2c\nu y_{n-2})}\mathbb{E}_{x_{n-1}}e^{\lambda(2x_{n-1}y_{n-2}+2c\nu x_{n-1}+x_{n-1}^2)}\right] \tag{54}$$

One can easily show that $\lambda(2y_{n-2} + 2c\nu + 1) \leq \lambda(2(n-2) + 2c\nu + 1) \leq \lambda(4n - 3) \leq \frac{\ln c}{1-c\nu}$, since $y_{n-2} = x_1 + \cdots + x_{n-2} \leq n - 2$. Therefore Corollary 2 implies $\mathbb{E}_{x_{n-1}}e^{\lambda(2x_{n-1}y_{n-2}+2c\nu x_{n-1}+x_{n-1}^2)} \leq e^{c\nu\lambda(2y_{n-2}+2c\nu+1)}$, since $x_{n-1}$ is independent with $y_{n-2}$. Plugging this into (54) leads to

$$\mathbb{E}_{y_{n-1}}\left[e^{\lambda(y_{n-1}^2+2c\nu y_{n-1})}\right] \leq \mathbb{E}_{y_{n-2}}\left[e^{\lambda(y_{n-2}^2+2c\nu y_{n-2})}e^{c\nu\lambda(2y_{n-2}+2c\nu+1)}\right] \tag{55}$$
$$= e^{c\nu\lambda(2c\nu+1)}\mathbb{E}_{y_{n-2}}\left[e^{\lambda(y_{n-2}^2+4c\nu y_{n-2})}\right] \tag{56}$$

By using the same arguments, we can show that

$$\mathbb{E}_{y_{n-1}}\left[e^{\lambda(y_{n-1}^2+2c\nu y_{n-1})}\right] \leq e^{c\nu\lambda(2c\nu+1)}e^{c\nu\lambda(4c\nu+1)}\mathbb{E}_{y_{n-3}}\left[e^{\lambda(y_{n-3}^2+6c\nu y_{n-3})}\right] \tag{57}$$
$$= e^{2c\nu\lambda(3c\nu+1)}\mathbb{E}_{y_{n-3}}\left[e^{\lambda(y_{n-3}^2+6c\nu y_{n-3})}\right] \tag{58}$$
$$...$$
$$\leq e^{c(n-2)\nu\lambda(c(n-1)\nu+1)}\mathbb{E}_{y_1}\left[e^{\lambda(y_1^2+2c(n-1)\nu y_1)}\right] \tag{59}$$

Note that $\mathbb{E}_{y_1}\left[e^{\lambda(y_1^2+2c(n-1)\nu y_1)}\right] = \mathbb{E}_{x_1}\left[e^{\lambda(x_1^2+2c(n-1)\nu x_1)}\right] \leq e^{c\nu\lambda(1+2c(n-1)\nu)}$, according to Corollary 2 and the fact that $\lambda(1 + 2c(n - 1)\nu) \leq \lambda(4n - 3) \leq \frac{\ln c}{1-c\nu}$. Combining this with (59), we obtain

$$\mathbb{E}_{y_{n-1}}\left[e^{\lambda(y_{n-1}^2+2c\nu y_{n-1})}\right] \leq e^{c(n-2)\nu\lambda(c(n-1)\nu+1)}e^{c\nu\lambda(1+2c(n-1)\nu)} = e^{c\nu\lambda(1+cn\nu)(n-1)} \tag{60}$$

By plugging this into (52), we obtain

$$\mathbb{E}_{y_n}e^{\lambda y_n^2} \leq e^{c\nu\lambda}e^{c\nu\lambda(1+cn\nu)(n-1)} = e^{c\nu\lambda((1+cn\nu)n-cn\nu)} \tag{61}$$
$$\leq e^{cn\nu(1+cn\nu)\lambda} \tag{62}$$

completing the proof. $\qquad\square$

### B.3 BINOMIAL AND MULTINOMIAL RANDOM VARIABLES

Next we analyze some properties of binomial random variables.

**Lemma B.5.** *Consider a binomial random variable $z$ with parameters $n \geq 1$ and $\nu \in [0, 1]$. For any $c \geq 1$ satisfying $c\nu \geq 1$ and any $\lambda \geq 0$, we have $\mathbb{E}e^{\lambda z^2} \leq e^{cn\nu(1+cn\nu)\lambda}$.*

*Proof.* Since $z$ is a binomial random variable, we can write $z = x_1 + \cdots + x_n$, where $x_1, ..., x_n$ are i.i.d. Bernoulli random variables with parameter $\nu$. Therefore applying Lemma B.2 completes the proof. $\square$

**Lemma B.6.** *Consider a binomial random variable $z$ with parameters $n \geq 1$ and $\nu \in [0, 1]$. For any $c \geq 1$ satisfying $c\nu < 1$ and any $\lambda \in [0, \frac{\ln c}{(1-c\nu)(4n-3)}]$, we have $\mathbb{E}e^{\lambda z^2} \leq e^{cn\nu(1+cn\nu)\lambda}$.*

*Proof.* Since $z$ is a binomial random variable, we can write $z = x_1 + \cdots + x_n$, where $x_1, ..., x_n$ are i.i.d. Bernoulli random variables with parameter $\nu$. Therefore applying Lemma B.3 completes the proof. $\square$

**Lemma B.7** (Multinomial variable). *Consider a multinomial random variable $(n_1, ..., n_K)$ with parameters $n$ and $(p_1, ..., p_K)$. For any $\delta > 0$:*

$$\Pr\left(\sum_{i=1}^{K} p_i^2 > \sum_{i=1}^{K}\left(\frac{n_i}{n}\right)^2 + 2\sqrt{\frac{2}{n}\ln\frac{K}{\delta}}\right) < \delta$$

*Proof.* Observe that

$$\sum_{i=1}^{K} p_i^2 - \sum_{i=1}^{K}\left(\frac{n_i}{n}\right)^2 = \sum_{i=1}^{K}\left[p_i^2 - \left(\frac{n_i}{n}\right)^2\right] \tag{63}$$

$$= \sum_{i=1}^{K}\left[p_i + \frac{n_i}{n}\right]\left[p_i - \frac{n_i}{n}\right] \tag{64}$$

$$= 2\sum_{i=1}^{K}\left(0.5p_i + \frac{0.5n_i}{n}\right)\left(p_i - \frac{n_i}{n}\right) \tag{65}$$

$$\leq 2\max_{i\in[K]}\left(p_i - \frac{n_i}{n}\right) \tag{66}$$

where the last inequality can be derived by using the fact that $\sum_{i=1}^{K}\left(0.5p_i + \frac{0.5n_i}{n}\right)\left(p_i - \frac{n_i}{n}\right)$ is a convex combination of the elements in $\{p_i - \frac{n_i}{n} : i \in [K]\}$, because of $1 = \sum_{i=1}^{K}\left(0.5p_i + \frac{0.5n_i}{n}\right)$. Furthermore, since $n_i$ is a binomial random variable with parameters $n$ and $p_i$, Lemma 5 in (Kawaguchi et al., 2022) shows that $\Pr\left(p_i - \frac{n_i}{n} > \sqrt{\frac{2p_i}{n}\ln\frac{K}{\delta}}\right) < \delta$ for all $i$. This immediately implies $\Pr\left(p_i - \frac{n_i}{n} > \sqrt{\frac{2}{n}\ln\frac{K}{\delta}}\right) < \delta$. Combining this fact with (66), we obtain $\Pr\left(\sum_{i=1}^{K} p_i^2 - \sum_{i=1}^{K}\left(\frac{n_i}{n}\right)^2 > 2\sqrt{\frac{2}{n}\ln\frac{K}{\delta}}\right) < \delta$, completing the proof. $\square$

## C    Experimental setup

More details about preprocessing and partition:

- We first preprocessed the images following Pytorch[3]:   The images are resized to $resize\_size = [256]$ using interpolation=InterpolationMode.BILINEAR, followed by a central crop of $crop\_size = [224]$. Finally the values are first rescaled to $[0.0, 1.0]$. Those operations are required for Pytorch pretrained models.
- For each run, we randomly choose 200 points in $[0.0, 1.0]^{C \times H \times W}$ to be the centroids, since each preprocessed image belongs to $[0.0, 1.0]^{C \times H \times W}$. Those centroids are used to build the small areas $\mathcal{Z}_i$ in the partition. Each training image $x$ will be assigned to area $\mathcal{Z}_i$ if it is closest to the centroid of $\mathcal{Z}_i$ amongst all centroids, according to the Euclidean distance.

## D    Additional experiment results

### D.1    Data geometries and partitioning strategies

We use the following **Mixture model (MM):** *each sample $(x, y)$ is generated by*

- *Randomly pick an index $z \sim Cat(\theta)$, a categorical distribution with parameter $\theta = (1/K, \ldots, 1/K) \in \mathbb{R}^K$*
- *Generate $x \sim \mathcal{N}(\mu_z, \nu)$, a normal distribution with mean $\mu_z = (0, \pi * z) \in \mathbb{R}^2$ and variance $\nu$*
- *Return class label $y = 1$ if $z$ is odd, and $y = 0$ otherwise.*

And three types of partitions $\Gamma$:

- ***T1:*** *A uniform grid partition that divides the data space into equally sized regions. However, this strategy may not align with the actual data distribution, potentially resulting in regions with highly imbalanced probability measures.*
- ***T2:*** *A partition formed by uniformly generating $K$ centroids to define the regions. Like T1, this method may not capture the underlying structure of the data.*
- ***T3:*** *A partition where the centroids $\mu_1, \ldots, \mu_K$ of the mixture components are fixed as region centers. This approach tends to yield more balanced regions, where $P(\mathcal{Z}_i) \approx P(\mathcal{Z}_j)$ for all $i, j$, for small variances.*

Figure 3 visualizes the mixture model (with $\nu = 1$) and those three partitions. This figure demonstrates that **T3** seems to best align with the data geometry, while the other two partitions can be much worse.

Figure 4 illustrates how the data geometry can be changed significantly when varying the variance in the mixture model. It is easy to see that the data-partition alignment quality can be entirely different even for the same partition.

### D.2    Comparation to existing generalization bounds

To further clarify the advantages of our bound, we carry out an additional comparison with robustness-based bounds developed by (Kawaguchi et al., 2022; Than et al., 2025), which are also model-dependent. In this comparison, we apply our bound under the mild setting used in Table 2, while we use $\delta = 0.05$ (corresponding to 95% confidence) and utilize the ImageNet validation set to approximate the intractable components for the bounds in (Kawaguchi et al., 2022; Than et al., 2025).

The results across 17 pretrained models are summarized in Table 4. The results suggest that our bound outperforms the existing robustness-based bounds in most cases, despite not relying on the validation set. This highlights the practical advantages and potential of our bound.

---

[3]`https://pytorch.org/vision/0.20/models/generated/torchvision.models.vit_b_16.html`

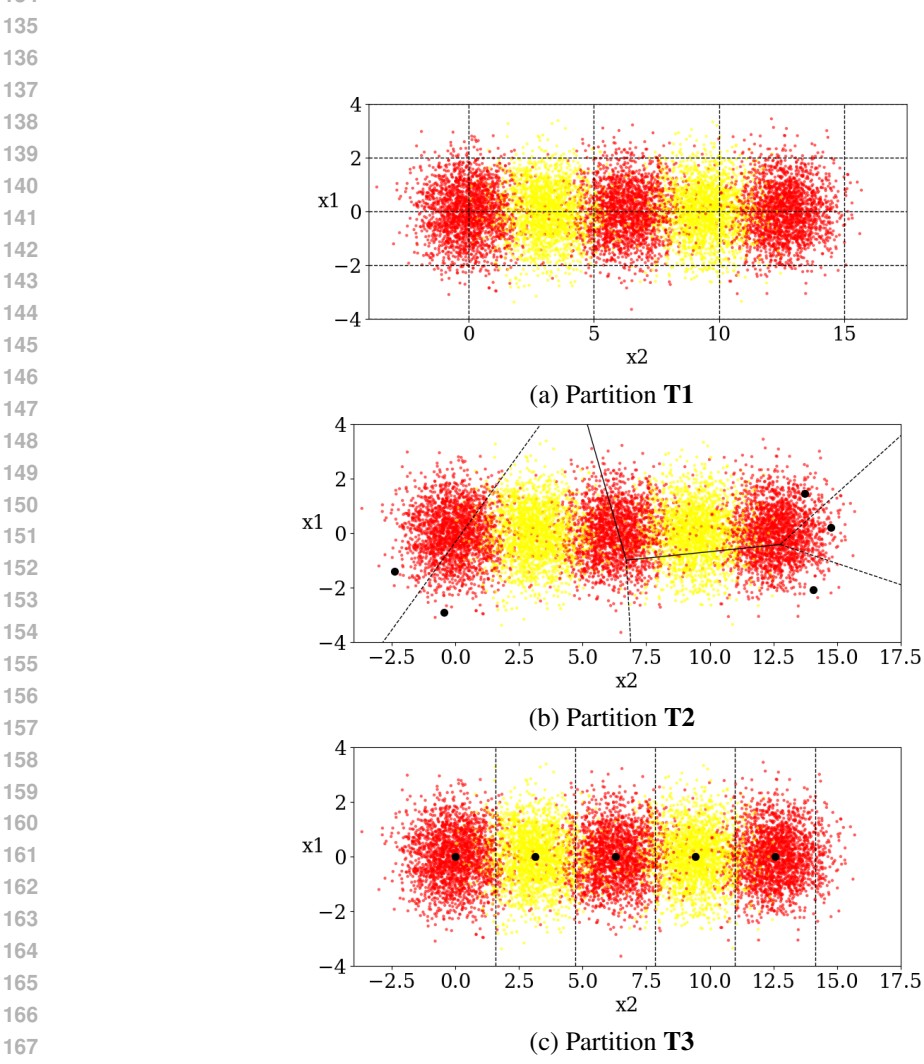

(a) Partition **T1**

(b) Partition **T2**

(c) Partition **T3**

Figure 3: Alignment between data distribution and partition.

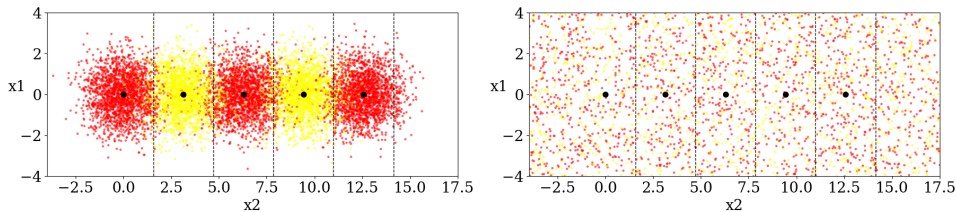

Figure 4: The mixture model for the cases: $\nu = 1$ (left), and $\nu = 10^4$ (right).

Table 4: Comparison with different model-dependent bounds for pretrained models on ImageNet. The prior bounds are approximated from the validation set, due to their intractability.

| Model | Test error | Bound (3) in (Kawaguchi et al., 2022) | Bound (8) in (Than et al., 2025) | Our bound (2) |
|---|---|---|---|---|
| ResNet18 V1 | 0.302 | 1.501 | 0.599 | 0.579 |
| ResNet34 V1 | 0.267 | 1.437 | 0.553 | 0.523 |
| ResNet50 V1 | 0.239 | 1.406 | 0.521 | 0.498 |
| ResNet101 V1 | 0.226 | 1.377 | 0.504 | 0.472 |
| ResNet152 V1 | 0.217 | 1.371 | 0.491 | 0.468 |
| SwinTransformer B | 0.164 | 1.323 | 0.432 | 0.431 |
| SwinTransformer T | 0.185 | 1.365 | 0.463 | 0.430 |
| SwinTransformer B V2 | 0.159 | 1.322 | 0.421 | 0.466 |
| SwinTransformer T V2 | 0.179 | 1.349 | 0.448 | 0.454 |
| VGG13 | 0.301 | 1.475 | 0.600 | 0.551 |
| VGG13 BN | 0.284 | 1.478 | 0.580 | 0.559 |
| VGG19 | 0.276 | 1.444 | 0.565 | 0.528 |
| VGG19 BN | 0.258 | 1.439 | 0.545 | 0.526 |
| DenseNet121 | 0.256 | 1.432 | 0.527 | 0.523 |
| DenseNet161 | 0.229 | 1.375 | 0.493 | 0.471 |
| DenseNet169 | 0.244 | 1.398 | 0.513 | 0.490 |
| DenseNet201 | 0.231 | 1.369 | 0.498 | 0.465 |

Table 5: Correlations to test error. $Unc(\Gamma)$ in Bound (1) is approximated from either the ImageNet *training set* or *validation set*. Bound (2) is computed from the training set alone.

| Quantity | Correlation to test error |
|---|---|
| Training error | 0.7899 |
| $Unc(\Gamma)$ (Train) | 0.7926 |
| $Unc(\Gamma)$ (Valid) | 0.9918 |
| Bound (2) | 0.7899 |
| Bound (1) | 0.9893 |

### D.3 CORRELATION BETWEEN TEST ERROR AND OUR BOUNDS

We investigate how well our bounds can correlate with test error. Our bounds contain two main parts: (1) Training error and (2) Uncertainty term $Unc(\Gamma)$. Due to being simplified from Bound (1), Bound (2) may not exhibit the full strength of our bounds in this work. Therefore, we take Bound (1) into consideration in this evaluation.

To examine the impact of the uncertainty term $Unc(\Gamma) = C\sqrt{\frac{u}{2n^2}\ln\frac{1}{\delta_1}} + g(\Gamma, \boldsymbol{h}, \delta_2)$ in Bound (1), we compute it from either the ImageNet training or validation set, using the mild setting for the paramteters. The results are reported in Table 5.

The results suggest that the uncertainty term in (1) captures meaningful characteristics of the trained models and correlate strongly with the test error. It also contributes a great role to the bounds, since Bound (1) exhibits a near-perfect correlation to test error. These highlight the practical relevance of our bounds for performance estimation.

### D.4 BETTER UNDERSTANDING OF GENERALIZATION

We next investigate how can theoretical bounds reflect the performance of a trained model and how predictive are the bounds? This is important when one wants to understand the main factors that lead to better performance/generalization of a model. It is also important to compare two specific/trained models of interest.

Table 6: Correlation between different components of Bound (1) with test error.

| Model | Test error | Align | Fair | Behavior |
|---|---|---|---|---|
| ResNet18 V1 | 0.30242 | 3.28869 | 64.6831 | 0.01444 |
| ResNet101 V1 | 0.22626 | 2.43343 | 46.0864 | 0.01066 |
| ResNet152 V2 | 0.17716 | 1.93093 | 38.8202 | 0.00850 |
| DenseNet201 | 0.23104 | 2.38285 | 45.8748 | 0.01045 |
| SwinTransformer B | 0.15888 | 1.72302 | 33.1667 | 0.00756 |
| VIT B 16 linear | 0.18114 | 2.14093 | 43.7663 | 0.00943 |
| **Correlation to test error** | | **0.98582** | **0.96397** | **0.98468** |

To this end, we take Bound (1) into consideration. Specifically, we focus on the following quantities for each model $h$ in the bound:

$$\text{Align}(h) = \sum_{i \in T} a_i(h)\sqrt{n_i/n} \tag{67}$$

$$\text{Fair}(h) = \sum_{i \in T} a_i(h) \tag{68}$$

$$\text{Behavior}(h) = \text{Align}(h) \cdot \sqrt{\frac{2\ln(2K/\delta_2)}{n}} + \text{Fair}(h) \cdot \frac{2\ln(2K/\delta_2)}{n} \tag{69}$$

Note that $Align(h)$ tells how well the model's local error can match with the data distribution. A better model should align better with the distribution's complexity, hence making $Align$ smaller. Meanwhile $Fair(h)$ tells the macro-level error of model $h$. It also suggests how fair for different local areas the model is. Finally, $Behavior(h)$ is the combined behavior, being an important part of Bound (1).

We use $K = 200, \delta_2 = 0.01$ and the ImageNet validation set to compute those quantities. The results for 6 pretrained models are reported in Table 6. We can observe that all of those quantities have extremely high correlations to the test error. $Align$ has the highest correlation, but $Fair$ has the lowest one. These results demonstrate that $Align$ can exhibit the quality of a model, and can be an accurate indicator for comparison between two models.

When visualizing the two lists $\{a_i : i \in T\}$ and $\{\sqrt{n_i/n} : i \in T\}$ in Figure 5, we observe a good correlation for some models, e.g., SwinTransformer B V2. Meanwhile ResNet18 V1 exhibited a much worse correlation. For areas with high probability mass (meaning large $\sqrt{n_i/n}$), those models often have small errors. However, those models have large errors on areas with very low probability density. The behavior in those areas (with probability mass $< 0.05$) seems quite noisy.

We further visualize those quantities $a_i$ and $\sqrt{n_i/n}$ for each local region in Figure 6. This visualization provides more details about the local behavior of a model, and supports further comparison between two models. For instance, while having comparable test error, ResNet152 V2 seems to slightly worse align with the distribution's complexity than VIT, specially for areas with low probability density.

*Remark* 3. The key insight from the result of this evaluation is that *the nature of the strong correlation in Table 6 reveals which geometric aspects of the data distribution and localized model behavior drive the generalization gap*. In particular:

- The decomposition in Bound (1) distinguishes *distributional concentration* from *local model stability*, two factors that classical norm-based or PAC-Bayes bounds cannot separate.
- The strong correlation between *the data-model behavior alignment* (as measured by $Align$) and the test error suggests that such an alignment may be a critical factor in generalization for modern large models.
- The local-loss terms indicate where the model allocates most of its residual error mass, providing a structured way to diagnose local weaknesses (e.g., regions with inconsistent predictions).

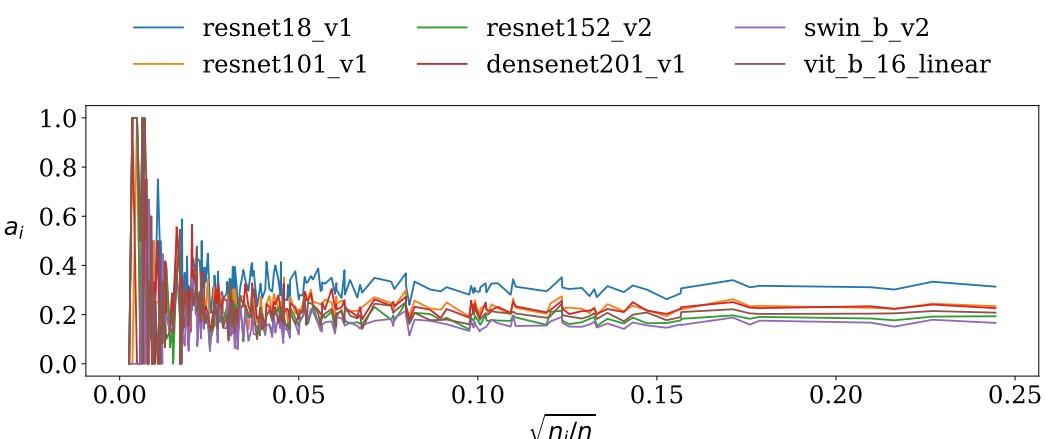

Figure 5: The alignment between local error with distribution's complexity. Note that quantities $\{\sqrt{n_i/n} : i \in T\}$ partly reflect the complexity of the data distribution. A high $\sqrt{n_i/n}$ means a high probability density in area $\mathcal{Z}_i$.

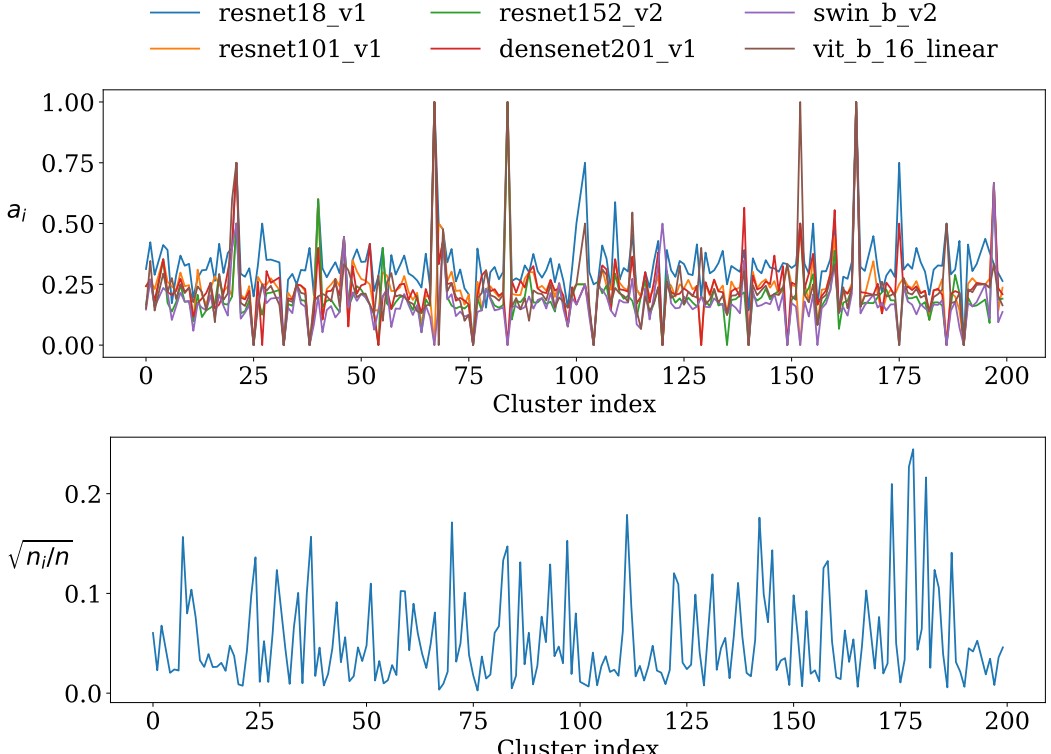

Figure 6: Distribution of local errors and samples on all local areas of the data space. Each cluster index represents an area.

Table 7: Results on Classification and Regression tasks.

(a) Classification results.

| Model | Test error | Train error | Our bound |
|---|---|---|---|
| Logistic Regression | 0.02467 | 0.01905 | 0.81254 |
| SVM | 0.02284 | 0.01271 | 0.78078 |
| XGBoost | 0.02638 | 0.01750 | 0.78557 |

(b) Regression results.

| Model | Test error | Train error | Our bound |
|---|---|---|---|
| Linear Regression | 0.14350 | 0.14187 | 1.06500 |
| SVR | 0.35403 | 0.35405 | 1.27718 |
| XGBoost | 0.10975 | 0.11070 | 1.03383 |

These observations open concrete avenues for future algorithms, e.g., geometry-aware sampling, region-wise curriculum learning, adaptive partition refinement, or regularizers designed to smooth local variations.

### D.5 COMPUTED BOUNDS FOR SMALLER OR SIMPLIER MODELS

In this section, we compute our bounds on several simple machine learning models for two tasks: classification and regression. Specifically, we train

- *three classifiers:* logistic regression, SVM, and XGBoost on a topic classification task using a news popularity dataset with 100K samples available at https://archive.ics.uci.edu/dataset/432/news+popularity+in+multiple+social+media+platforms.
- *three regression models:* linear regression, support vector regression (SVR), and XGBoost, using the VirusShare dataset with 107K samples available at https://archive.ics.uci.edu/dataset/413/dynamic+features+of+virusshare+executables.

For all experiments, we split each dataset into training set and testing set with ratio 8:2 and compute our bound using the setting $K = 200$, $\delta_2 = 0.01$, $\delta_1 = 0.04$, $\alpha = 100$, as in other experiments. For the classification tasks, we employ the 0-1 loss to measure both training and test error, building the partition by clustering on the TFIDF vector space with random centroids initialized. For the regression tasks, we use the L1 loss for computing the training and test loss, building the partition by clustering on the standard-scaled vector space with random centroids initialized.

The evaluation results are reported in Table 7. We observe that our bound is non-vacuous for all the classifiers. For this experiments, our bound seem to be high and far from the test error. The main reason may come from the small size of the datasets (about 100K samples vs. 1.2M ImageNet images). Some other reasons may be the large value of $K = 200$ and the misalignment between the partition and the data distribution.

