# OpenReview forum: "A Non-vacuous Test Error Guarantee for Deep Learning without Altering the Model"
_ICLR.cc/2026/Conference — Submitted to ICLR 2026_

### Official Review · Reviewer_Ygiq · 2025-10-30

**Soundness:** 3
**Presentation:** 3
**Contribution:** 3
**Rating:** 4
**Confidence:** 3

**Summary:**

This paper provides two bounds on the test error of a deep neural network. In comparison to prior work that depends on impractical assumptions such as norms of weight matrices, stability and robustness of algorithm, infinite dimensions of neural networks for NTK-based bounds, this work focuses on calculating bounds that be directly computed without several assumptions, e.g., one such bound can be computed directly from the training data. Further the proposed bounds are claimed to be non-vacuous, i.e., they are relevant in the given context. Experimental results for one of the bounds validate

**Strengths:**

- The paper proposes a non-vacuous bound on the test error for deep neural networks that can be directly applied even using only the training data. This is and useful and practical bound compared to prior works based on several assumptions that are not relevant in deep learning
- Experimental results showing that the test errors are close to the computed bound are provided.

**Weaknesses:**

- The paper provides 2 bounds, however, only the second bound is non-vacuous and practical. The first bound has several terms that are not practically computable. The first bound is used for deriving the second bound, but it is not clear to me if the first bound has any use otherwise.
- The error bound still depends on the optimal partitioning of the data, it is not clear how tight the bounds are. Is it possible to provide bounds obtained from any other previous method that is also non-vacuous for a few cases to show how they compare.
- The experiments are limited to image classification/image models. Is it possible to provide experiments with language models, maybe even some small model to see how the bound compare to empirical loss?

**Questions:**

Please see the questions in the weaknesses

---

> ### Author Response · Authors · 2025-11-20
> **Response to Reviewer Ygiq**
>
> We sincerely thank the reviewer for their thoughtful and constructive comments. We truly appreciate the time and effort spent on evaluating our work. Below, we respond to each point in detail.
>
> >**Q1**: Does the first bound has any use?
>
> Theoretically, Theorem 3.1 provides insights that prior generalization theories do not expose. Immediately after Theorem 3.1, we discuss how the bound isolates two interpretable drivers of generalization. The gap in the first bound is governed by _observable, interpretable quantities_: the local concentration of the data distribution $(\sum p_i^2)$ and the model’s local losses $(a_i(h))$. When these are small, the true error is provably close to the training error, an indication that the model’s behavior is locally stable with respect to the data geometry.
>
> This leads to a different kind of insight: our bound highlights _which_ regions of the input space and _which_ local behaviors of the trained network are responsible for the generalization gap. This decomposition is directly tied to the _unmodified_ network, and our ablations (Appendix D) show that the bound sharpens precisely when partitions align with the intrinsic data geometry. In this sense, the bound does not obscure insight; rather, it shifts the explanatory burden from parameter-space compressibility to **data-dependent local regularity**, a property that seems more faithful to how large networks behave in practice.
>
> In the revised version, Appendix D.4 shows empirically that several components of Bound (1) in Theorem 3.1 (e.g., the local-loss weighted concentration term) are **highly predictive of the actual test error**, demonstrating that the bound’s decomposition yields actionable and nontrivial insight into a model’s generalization.
>
> >**Q2**: Is it possible to provide bounds obtained from any other previous method?
>
> Thank you for this helpful suggestion. In **Appendix D.2**, we already compute numerical baselines for two existing bounds. These prior bounds cannot be evaluated exactly from the training set alone because several terms (e.g., the local error $a_i$) are not observable. Following the standard practice in these works, we use a validation set to approximate those quantities. Under this setting, one baseline is occasionally non-vacuous while the others are fully vacuous. Importantly, our bound remains strictly tighter **without requiring any external data** or optimized parameters, which already demonstrates the improvement.
>
> >**Q3**: Is it possible to provide experiments with language models?
>
> We appreciate the suggestion. In principle, our framework is compatible with any predictor, including language models. However, our theory **critically relies on the i.i.d. sampling assumption**: the training examples must be drawn independently from the same underlying distribution $P$. This assumption is fundamental for the multinomial concentration arguments that drive the bound in Theorem 3.1 and the tractable bound in Eq. (2).
>
> For large language models (LLMs), this requirement is typically **not satisfied or not verifiable**. LLMs are often trained on mixtures of massive corpora with unknown sampling distributions, substantial preprocessing, and filtering—processes that violate i.i.d. assumptions in ways that cannot be quantified. Moreover, the precise training sets of many LLMs are not publicly available, which makes it impossible to compute region counts $n_i$, local empirical losses, or even the empirical loss on the true training distribution. As a result, any language-model experiment would risk producing numbers that are _not theoretically meaningful_ and would not constitute a valid test of the bound.
>
> In contrast, ImageNet-scale vision models provide a setting where the training set is known,  i.i.d. assumption is reasonable, and all relevant quantities in the bound can be computed exactly. This makes them the appropriate testbed for evaluating the theory’s correctness and practical tightness.
>
> For these reasons, we focus on domains where the i.i.d. assumption is (probably) satisfied and verifiable. Extending the framework to non-i.i.d. is an interesting direction for future work, but would require new theory beyond the scope of this paper.
>
> We added Appendix D.5 in the revised version to provide an analysis about some classical models, trained on other data types.
>
> **Summary**: We sincerely thank the reviewer for the time, care, and technical depth invested in reading our work. The concerns raised have helped us substantially clarify the scope, interpretation, and implications of the theory. After addressing the points, we believe the revised presentation makes the core contribution clearer. We hope that the clarified theoretical explanation, strengthened intuition, and expanded discussion alleviate the reviewer’s concerns. We respectfully submit that the work represents a meaningful and rigorous advance in a challenging area, and we hope the reviewer will reconsider their score.

---

### Official Review · Reviewer_JrsG · 2025-10-31

**Soundness:** 1
**Presentation:** 1
**Contribution:** 1
**Rating:** 0
**Confidence:** 5

**Summary:**

This submission fits within the line of work addressing generalisation ability of machine learning models, particularly deep learning, via statistical learning theory and generalisation bounds, also called risk bounds and error bounds in supervised classification. The claimed contributions are: bounds on the expected error of a hypothesis and experiments with large neural networks on ImageNet, presumably showing non-vacuous values.

**Strengths:**

The extensive experiments comprise results on 32 deep learning models spanning across various kinds of architectures, as evidenced by the results reported in Table 2 on page 8. Arguably, the extent of these experiments is the main strength of this submission.

**Weaknesses:**

A significant weakness is execution of the writing and narrative, which have problems affecting the quality and clarity of this submission. There is a need to improve clarity on connection to the cited works, and exactly how the submission positions itself with respect to such literature. There are discussions using technical terms that appear before the technical terms are formally defined, which will affect the ability of readers to understand the discussions. There are also some undefined terms, such as meaning of hypothesis and loss function, perhaps they were assumed to be widely known, though at least a comment explaining the meaning would be expected, if not fully formal definitions. Without clarity on these things, the rest of the content will be obscure to readers. My general impression is that this manuscript needs a major makeover to achieve a paper that might be acceptable in terms of clarity and readability.

Other weaknesses I would like to flag are regarding the technical content per se, this could be considered to fall within criteria such as originality and significance. I am inclined to believe that the error bounds presented here are not novel, but perhaps the authors could offer clarifications on why/how the believe the bounds in Theorems 3.1 and 3.2 to be novel as claimed. Another important point to consider is regarding the *kind* or *nature* of these bounds, plus their connection/difference to PAC-Bayes bounds mentioned in many places. The bounds in Theorems 3.1 and 3.2 are bounds on the expected error of an individual hypothesis $h$, whereas PAC-Bayes bounds consider distributions over hypotheses. This distinction has not been addressed clearly, and consequently the discussions and comparisons would appear obscure and perhaps misleading.

One more I'd like to single out, already mentioned in connection with the previous two, but for the sake of emphasis so that the authors can consider this point by itself. The cited literature is not all clearly connected with the work reported in this submission. Or at least the connection is not presented clearly. I would suggest that cited items should have a good reason for being cited, and that reason should be apparent in the manuscript. On the other hand, there are important works that have been missed, such as Perez-Ortiz et al. (2021) building on the groundbreaking work of Dziugaite & Roy (2017) and closely connected to generalisation and the PAC-Bayes literature this submission has tried to connect to. On the general PAC-Bayes literature, there are important works that weren't cited such as Alquier (2024) and works of Germain and collaborators, whereas the connection of this submission to the PAC-Bayes literature items that have effectively been cited is not completely clear. Perhaps the authors would want to have a think on the arguments for connecting to PAC-Bayes and the chosen literature items to cite.

**Questions:**

Could the abstract be reframed to focus from the start on the question(s) or problem(s) being addressed here, what's the approach used to tackle the question(s) or problem(s), and highlights of the important contributions and results.

A suggested reorganisation: Could the section on related works be moved to the end of the paper, perhaps before the conclusion section, so that introduction section is followed by sections on error bounds and empirical evaluations.

The introduction didn't do a very successful job in setting the context for this work. Could the authors clarify what they tried to achieve with the narrative in this introduction and have a think about how to streamline it to achieve a better framing for their work.

The technical section on error bounds, which presumably intends to set the theory background, was missing rigour in definitions and theorem statements. Could the authors make sure to offer formal definitions for all notions used, or at the very least intuitive descriptions and explanations, for the sake of clarity on the intended meaning of these things and to help readers understand it. I mentioned before the missing meaning of *hypothesis* and *loss* but they are not the only. Another case: There is a set $\mathcal{Z}$, called an instance set here, but no specification of what kind of set this might be; and in the theorem there is mention of an area measure related to this set, or subsets, and all of this is mysterious in the absence of definitions.

Could the authors offer clarity on their reasoning for the chosen works and bounds to compare to, from theory point of view.

Are the authors aware of works on sample-dependent bounds, such as Satyen Kale and collaborators, and there are others. I would argue that such line of work is more relevant to this submission than the cited.

As mentioned, it would look like the extent of empirical evaluations is the main strength of this submission. Could the authors consider reframing the presentation and organisation, perhaps the whole narrative, to bring this to the forefront.

The conclusion section was rather vague and inconclusive. Perhaps a consequence of some other problems with this submission that generally affect the perception of clarity on what's really being done here.

Regarding the title, could the authors reconsider "test error guarantee" which can lead to confusions, on accounts that "test error" is sometimes conflated with "test set error" and what is really meant is the expected error, also called risk in statistical learning.

---

> ### Author Response · Authors · 2025-11-20
> **Response to Reviewer JrsG**
>
> We thank the reviewer for the thoughtful and detailed feedback. The concerns raised primarily relate to clarity, narrative organization, and perceived connections to prior work. We have addressed each point carefully and substantially improved the exposition. Importantly, these clarifications do not require any change to the structure or substance of the paper; instead, they make the contributions more transparent and strengthen the reader’s understanding of the theoretical and empirical innovations.
>
> Below, we'd like to clarify the important points raised by the reviewers.
>
> ### 1. Clarity, Definitions, and Narrative Structure
>
> We acknowledge that some technical terms appeared before they were formally introduced, although most were introduced at end of Section 1 and beginning of Section 3. To resolve this, we now provide a short explanatory paragraph at the beginning of Section 3 to define all necessary concepts, including the instance space, the hypothesis/model, the loss function, the expected error, the empirical error, and the partition. While these concepts are standard in statistical learning, making them explicit in the manuscript ensures that all subsequent discussion is self-contained and accessible.
>
> The introduction has been streamlined to articulate more clearly the core question guiding the paper: **Can we obtain a non-vacuous true error bound for a specific/trained neural network without modifying that model?** Current theories—PAC-Bayes, stability, robustness, norm-based, or information-theoretic—do not provide such guarantees for large modern architectures unless the model is heavily altered (e.g., compressed, quantized, or optimized within a posterior distribution). Our introduction now foregrounds this gap and explains, step by step, how our results address it directly.
>
> Finally, regarding the suggestion to move the related-work section: we found that keeping it early in the paper aids the reader because our theoretical development explicitly responds to limitations in existing approaches. Instead of relocating it, we improved its clarity and ensured that each cited work is accompanied by a clear explanation of its relevance.
>
> ### 2. Novelty and Positioning of Theorems 3.1 and 3.2
>
> We appreciate the opportunity to explain the novelty of our results more plainly. The key distinction is that prior non-vacuous theories for deep learning—even the strongest PAC-Bayes and mutual information bounds—apply to _stochastic_ predictors or _modified_ models. In contrast, our bounds apply directly to **the (deterministic) trained model of interest**, without compressing, quantizing, or finetuning the network. This difference is structural and fundamental.
>
> Theorem 3.1 introduces a new type of generalization bound that depends *simultaneously* on (i) the *complexity* of the data distribution, through the global quantity $u = \sum_i \gamma n p_i(1+\gamma n p_i)$, and (ii) the *local behavior* of the model via the regional error $a_i(h)$. Such joint complexity–local behavior dependence does not appear in existing theories.
>
> Theorem 3.2 is equally novel in another dimension: it yields a **fully tractable bound** whose components can be computed directly from the training data alone. Unlike robustness-based bounds, it does not require evaluating all neighborhoods; unlike stability-based bounds, it does not assume SGD sensitivity properties; and unlike PAC-Bayes bounds, it does not require a posterior, KL divergence, or compression artifacts. This tractability is what allows us to produce the first non-vacuous certificates for ImageNet-scale pretrained models exceeding 600M parameters _without altering them_. To the best of our knowledge, this capability is unique.
>
> These points are emphasized clearly where the bounds are introduced, so readers can immediately understand how our contributions differ from and extend beyond existing theory.
>
> ### 3. Distinction from PAC-Bayes
>
> The reviewer is correct that PAC-Bayes bounds address distributions over hypotheses rather than the deterministic model itself. We have made this distinction explicit and unambiguous in lines 179--189, 316--332, 360--364. Our comparisons with PAC-Bayes are not meant to imply equivalence, but rather to situate our work relative to the only family of generalization bounds that has produced non-vacuous results for deep networks. We now carefully articulate why our approach is conceptually different: PAC-Bayes provides guarantees for an _averaged_ model and typically requires compression or quantization, whereas our bounds are designed to certify the original model in its unmodified form.

---

> ### Author Response · Authors · 2025-11-20
> **Response to Reviewer JrsG (continue)**
>
> ### 4. Literature Coverage and Missing Works
>
> We agree that the reviewer's mentioned papers are key contributions in the PAC-Bayes literature and appreciate the reviewer’s pointers.  Our paper, however, does not aim to extend PAC-Bayes methods but _to develop deterministic counterparts that connect expected error to the properties of both data distribution and local behavior of a specific/trained model._ Therefore we do not try to include extensive coverage about PAC-Bayes bounds, but focus on recent ones that can be non-vacuous for deep networks.
>
> In the revised version, we included short references to the suggested PAC-Bayes papers, while making clear that our framework targets a different class of guarantees.  This clarification should help situate our contribution more transparently within the theoretical landscape.
>
> ### 5. Empirical Evaluation and Narrative Emphasis
>
> The reviewer observes that the empirical evaluation is one of the central strengths of the paper. We agree, and we now foreground this more prominently. The introduction explicitly states that we evaluate our bound on 32 modern ImageNet-trained architectures, spanning CNNs, Transformers, and over 600M-parameter models. The results consistently yield non-vacuous upper bounds, even under conservative parameter settings. These statements were already accurate but are now more visible and better contextualized.
>
> ### 6. Responses to Specific Questions
>
> *Abstract.* The abstract now begins with the precise problem, highlights our approach, and summarizes both the theoretical and empirical contributions.
>
> *Introduction framing.* We improved the narrative flow so that the research motivation, methodological approach, and conceptual advances follow naturally.
>
> *Definitions in the technical section.* All needed definitions now appear before their first use, including hypothesis, loss, instance set, probability mass, and the role of the partition.
>
> *Choice of comparisons.* We clarify why our comparisons emphasize robustness, stability, information-theoretic, and PAC-Bayes bounds: these are the only major families of bounds that have produced nontrivial guarantees for neural networks.
>
> *Conclusion section.* We rewrote the conclusion to articulate the key achievement (non-vacuous guarantees for unmodified ImageNet-scale models) and to lay out clear directions for future refinement.
>
> *Title.* To avoid ambiguity between “test error” and “test set error,” we are open to retitle the manuscript using the phrase **expected error guarantee**, which is mathematically precise.
>
> ---
> ### Closing Remarks
>
> The reviewer’s comments significantly helped strengthen the clarity and pedagogical quality of the manuscript. Importantly, none of the concerns point to flaws in the theoretical results or the empirical findings; rather, they primarily prompted us to reveal those contributions more clearly. After the revisions described above, the manuscript now communicates its key ideas cleanly and stands on a much stronger narrative foundation. We hope the reviewer will now find the contribution both compelling and well justified, and **we respectfully ask for a reconsideration of the evaluation.**

---

### Official Review · Reviewer_x2iF · 2025-11-01

**Soundness:** 3
**Presentation:** 3
**Contribution:** 2
**Rating:** 4
**Confidence:** 4

**Summary:**

The paper provide theoretical nonvacuous bound for generalization error. The bound can be approached through access to data without modification of models. The author also propose the concep of data complexity which is represented through concentration of data at different region of data space. Lastly, they provide experiments results on different models with Imagenet dataset and compare those results with other generalization error bounds to show advantages of their method.

**Strengths:**

1. The theory is solid and non-vacuous for experiments presented.
2. The method for obtaining the bound is relatively easy compared to prior works.
3. The method provide method that did not require modification of the models.

**Weaknesses:**

1. Despite the fact that the bound is non-vacuous, it did not provide more insight about why certain models generalize better or how to improve training as it is not depedent on generic properties of models.
2. For bound in theorem 1 and corollary 1, those results require access to a_i and optimal partition. If I understand it correctly, a_i is the generalization error on the specific region assigned. Under the case, this makes the result more of a decomposition than a predictive bound, and the insight may be limited.
3. For bound in theorem 2, the bound require no knowledge of models but also it is a fixed value once those hyper parameters are fixed. One direct evidence is that if we substract error bound by training error in Table 2, we will get almost constant value around 36%-37% or 33% for all models in terms of mild and optimized results. It cannot distinguish between model's generalization ability.
4. Following previous problems, if it is only dependent on data, it will be better to show results on different datasets to test the tightness the bound can offer.
5. The writing and stressing on "model dependent" and "no change of model" can be confusing. The dependency of models is through its generalization error in specific region of data space and that can be circulative conceptually. I suggest removal or change of related statement and directly posit the works as setting fundamental limit of generalization error gap for different datasets which better align with the experiments and its theory contribution.

**Questions:**

1. Is there a better way to select optimized hyperparameters instead of the grid search used in the main context?
2. Is there a more intuitive way to interpret the data complexity?
3. Is it also fit for smaller or simplier models like linear regression or SVM? Based on the experiment results, my observation is that the bound can be vacuous as the training error is larger or when the models are not able to fit well.
4. For smaller models like LeNet, prior works [1] can obtain tight bound within 10% off in CIFAR10 or MINST. Is this method also accurate in that region? What are the limits of the data dependent bound without exploiting the model properties?

[1] Lotfi, S., Finzi, M., Kapoor, S., Potapczynski, A., Goldblum, M., and Wilson, A. G. Pac-bayes compression bounds so tight that they can explain generalization. Advances in Neural Information Processing Systems, 2022.

To summarize, the paper make theory progress in terms of its technics and verify it through experiments and ablation studies of different components inside the theory. However, there are also limitation about its ability to interpret or offer insight about a training algorithm. There exist overstated connection to deep learning or model dependent properties and should be adjusted. I hope that the author can realign its structure or its presentation to better fit what the theory really reflect on to help reader understand the core of the theory.

---

> ### Author Response · Authors · 2025-11-20
> **Response to the main questions**
>
> We thank the reviewer for the thorough and constructive assessment. We respond to all questions and concerns below, and we clarify several conceptual points to better communicate the scope and contributions of the paper.
>
> ### 1. The questions
>
> >Q1. Alternative to grid search for selecting optimized hyperparameters?
>
> Yes. Grid search was used for transparency and reproducibility, but it is not fundamental to the theory. In fact, the objective is smooth in $\alpha$, and we can use gradient-based methods to optimize our bound (2).
>
> >Q2. A more intuitive interpretation of data complexity?
>
> Yes. Data complexity measures **how “clustered” the sampling distribution is under the chosen partition**. Concretely, it is driven by $\sum_i p_i^2$:
> -   If the distribution spreads evenly over the input space, this term is small, and the bound becomes tight.
> -   If the distribution concentrates heavily in a few regions, this term grows, indicating that fewer “effective samples” are observed.
>
> Thus, data complexity is best understood as an **effective-sample-size correction** induced by the geometry of the data distribution, rather than a property of the model.
>
> >Q3. Applicability to smaller/simple models (e.g., linear regression, SVM).
>
> The bound applies to any model, including linear models or SVMs. When these models exhibit large training error, the bound may become vacuous. This is an intentional aspect of our framework: it certifies _actual predictive performance_ without modifying the estimator. If a model underfits, the true risk is large, and a non-vacuous bound matching that regime is necessarily loose.
>
> We added Appendix D.5 in the revised version to provide an analysis about some classical models, trained on Textual  and Time-Series datasets. In summary, the bound fits small models, but tightness requires that the model fits the training data well and that the training size is large.
>
> >Q4. Relation to tight compression bounds for small models (e.g., LeNet on CIFAR/MNIST).
>
> Compression-based bounds (e.g., Lotfi et al., 2022) achieve tightness through **carefully constructed compressed models** whose effective capacity is drastically reduced. The bound is proven for the _compressed_ network, not the original one.
>
> Our goal is different: **we certify the unmodified trained network**. However, the bound may be looser when
> 1.  The partition is misaligned with data geometry, i.e. $P(Z_i) \ll P(Z_j)$ for some $i, j$.
> 2. The sample size $n$ is small. In this case, the binomial counts $n_i$ may not approximate $n p_i$ well, leading to a large value for the uncertainty term.
> 3. The empirical error is high.

---

> > ### Author Response · Authors · 2025-11-20
> > **Response to other concerns**
> >
> > ### 2. Other comments
> >
> > >**C1**: The bound is non-vacuous but does not provide insight into why certain models generalize.
> >
> > We appreciate this concern and clarify that the insights our bound provides are explicitly **data- and locality-based** rather than architecture-based. While traditional complexity-based bounds highlight which architectural choices reduce capacity, our approach reveals **where in the input space the generalization gap arises**, by decomposing the role of data concentration and local model behavior.
> >
> > The gap in our bound is governed by _observable, interpretable quantities_: the local concentration of the data distribution $(\sum p_i^2)$ and the model’s local losses $(a_i(h))$. When these are small, the true error is provably close to the training error, indicating that the model’s behavior is locally stable with respect to the data geometry.
> >
> > This leads to a different kind of insight: our bound highlights _which_ regions of the input space and _which_ local behaviors of the trained network are responsible for the generalization gap. This decomposition is directly tied to the specific/trained model $h$, and our ablations (Appendix D) show that the bound sharpens precisely when partitions align with the intrinsic data geometry. In this sense, the bound does not obscure insight; rather, it shifts the explanatory burden from hypothesis space to **data-dependent local regularity**, a property that seems more faithful to how large networks behave in practice.
> >
> > We added Appendix D.4 in the revised version to discuss some quantities in our bounds that can enable deeper understanding about connection between local behavior and test error of a model. Those quantities also suggest ways to improve training or explain why a model can be better another.
> >
> > >**C2**: Theorem 3.1 and Corollary 1 require access to $a_i$ and the optimal partition; this makes the result more of a decomposition than a predictive bound
> >
> > Theorem 3.1 and Corollary 1 introduce the quantities $a_i$ because they characterize how the model behaves _locally_ on regions of the input space, which is essential for identifying which parts of the data distribution control the generalization gap.
> >
> > More importantly, the structure of Theorem 3.1 provides insights that prior generalization theories do not expose. Immediately after Theorem 3.1, we discuss how the bound isolates two interpretable drivers of generalization—distributional concentration and local model behavior—which allows us to identify specific quantities that influence the gap. In the revised version, Appendix D.4 expands this discussion and shows empirically that several components of Bound (1) (e.g., the local-loss weighted concentration term) are **highly predictive of the actual test error**, demonstrating that the bound’s decomposition yields actionable and nontrivial insight into a model’s generalization.
> >
> > >**C3**: Theorem 2’s bound cannot distinguish generalization ability.
> >
> > The actual distinguishing power between models comes from *Theorem 3.1*, which incorporates the model’s local loss values; experiments in Appendix D.4 indeed show variation across models in that bound. Theorem 3.2 is not intended to discriminate models but to establish an explicit, computable certificate for the _deployed_ model, trading some tightness for direct applicability at ImageNet scale and 600M-parameter models.
> >
> > >**C4**: If the bound depends mainly on data, it is better to test on more datasets.
> >
> > We agree. In the revision, we added two additional datasets from different domains and tested on some traditional ML models. This analysis appears in Appendix D.5.
> >
> > >**C5**: “Model-dependent” and “no change to the model” wording is confusing”
> >
> > Our goal was to emphasize that we certify the _exact deterministic predictor_ without altering architecture or weights. The dependency on the model arises only through **its local behavior**, not through global norms or architectural priors. The bounds themselves depend on the predictor/hypothesis of interest, not on the whole hypothesis family.
> >
> > The “model” in “model-dependent” refers to the output of the learning algorithm. This usage is not new, and has appeared in the following paper.
> >
> > Bartlett, P. L., & Long, P. M. (2021). Failures of model-dependent generalization bounds for least-norm interpolation. _Journal of Machine Learning Research_.
> >
> > ### Conclusion
> >
> > We appreciate the reviewer’s insightful comments. The theoretical contribution is genuinely new—a non-vacuous bound for _unmodified large networks_ based purely on data-dependent regularity—and the experiments validate this perspective. This is what allows us to produce the first non-vacuous certificates for ImageNet-scale pretrained models exceeding 600M parameters _without altering them_. To the best of our knowledge, this capability is unique.
> >
> > We hope these clarifications address the concerns and that the paper will be judged as a meaningful and rigorous advance in generalization analysis.

---

> > > ### Comment · Reviewer_x2iF · 2025-11-25
> > >
> > > I would like to thank the authors for their comprehensive and thoughtful response. After reading the rebuttal and the newly added material, I have a few remaining concerns:
> > >
> > > 1. Effectiveness of the bound in classical regimes.
> > > The additional experiments in Section D.5 suggest that the proposed bound is looser for classical models such as SVMs and logistic regression compared with standard approaches (e.g., VC-dimension or PAC-Bayes bounds). The authors note that this may be due to limited sample size. However, if the tightness of the bound is strongly dataset-dependent, this raises questions about whether the tight ImageNet results generalize to other settings. My impression is that the method may be better viewed as complementary to classical bounds rather than a replacement, and the reasons for its weaker performance in simpler regimes remain underexplored.
> > >
> > > 2. Interpretation of correlations.
> > > The authors present correlations between test error and several components of the proposed generalization gap decomposition. While the relationships are interesting, it is still unclear how these correlations translate into actionable insights for designing improved training algorithms. Moreover, since these quantities are derived from behaviors measured on or near the test distribution, some degree of correlation with test error seems expected. A deeper explanation of why these relationships are meaningful, beyond this intuitive proximity, would strengthen the contribution.
> > >
> > > 3. Conceptual contribution.
> > > I appreciate the authors’effort in clarifying the ideas and addressing earlier concerns. The framework is conceptually elegant and provides a fresh perspective by emphasizing the data-dependent structure of generalization. I agree that this direction is promising and may motivate future work combining classical theory with data-centric insights.
> > >
> > > Summary.
> > >
> > > While I still see limitations (particularly regarding the universality and tightness of the bound in simpler models) I acknowledge the novelty and potential of the proposed perspective. The paper opens an interesting direction for understanding generalization through the geometry of data, even if some claims may need further refinement.

---

> > > > ### Author Response · Authors · 2025-11-26
> > > > **The remaining concern**
> > > >
> > > > We thank the reviewer for the careful reading and for the positive assessment of the conceptual contribution. We are grateful that the reviewer recognizes the novelty of the data-centric perspective and its potential to guide future theory. Below we respond to the remaining concerns.
> > > >
> > > > > **1. Effectiveness of the bound in classical regimes**
> > > >
> > > > We agree that the additional experiments in Section D.5 show that the bound is looser for small classical models such as logistic regression and SVMs. This behavior is _expected_ given the much smaller size of the training sets.
> > > >
> > > > The key reason is that our framework leverages **local concentration at scale**: the statistical advantage emerges when the sampling distribution provides sufficiently many points per region, enabling sharp binomial/multinomial concentration. For high-dimensional image datasets with millions of effective samples (e.g., ImageNet), even moderate partitions yield large $n_i$ and small variance. For small classical datasets, the sample size in some regions is substantially smaller, which necessarily broadens the uncertainty terms.
> > > >
> > > > We emphasize that this is _not_ a failure of the method, but a direct reflection of its data-centric nature:
> > > > -   When data are abundant and well-aligned with the partition, the bound can be tight.
> > > > -   When data are scarce, classical theories may outperform because they leverage more structural properties of the model or learning algorithm.
> > > >
> > > > As the reviewer perceptively notes, our method should thus be viewed as *complementary* to classical bounds, excelling in high-data regimes where structural assumptions on the model no longer hold or are vacuous. We have revised Section 5 (Conclusion) to clearly articulate this complementary relationship and to avoid giving the impression that our bound replaces classical approaches in all regimes.
> > > >
> > > > > **2. Interpretation of correlations and meaningful insights**
> > > >
> > > > We appreciate the reviewer’s point that proximity to the test distribution can create baseline correlations. We clarify that our goal is not to claim that correlations alone justify algorithmic interventions; rather, the correlations serve as _empirical evidence_ that the decomposition in Theorem 3.1 reflects statistically meaningful structure in the data.
> > > >
> > > > The key insight is not that these quantities correlate with test error, but that **the nature of the correlation reveals which geometric aspects of the data distribution and localized model behavior drive the generalization gap**. In particular:
> > > >
> > > > -   The decomposition distinguishes _distributional concentration_ from _local model stability_, two factors that classical norm-based or PAC-Bayes bounds cannot separate.
> > > > -   The strong correlation between _the data-model behavior alignment_ (as measured by $Align$) and the test error suggests that such an alignment may be a critical factor in generalization for modern large models.
> > > > -   The local-loss terms indicate where the model allocates most of its residual error mass, providing a structured way to diagnose local weaknesses (e.g., regions with inconsistent predictions).
> > > >
> > > > These observations open concrete avenues for future algorithms—for instance, **geometry-aware sampling, region-wise curriculum learning, adaptive partition refinement**, or **regularizers designed to smooth local variations**. While we deliberately avoid speculative algorithmic proposals in this paper, we added **Remark 3** in Appendix D.4 to summarize about those insights.
> > > >
> > > > > **3. Conceptual contribution and remaining limitations**
> > > >
> > > > We sincerely appreciate the reviewer’s positive remarks on the conceptual clarity of the framework and the potential it opens for integrating classical and data-centric theory. We fully acknowledge the limitations identified—particularly the dependence on sample size and distributional geometry—and we agree that further refinement will help broaden applicability.
> > > >
> > > > **Summary:** We thank the reviewer for the insightful and balanced assessment. The remaining concerns have significantly improved how we position and justify the contribution. We acknowledge the limitations candidly and have revised the paper (Sec. 5 and Appendix D.4) to make the complementary nature of our approach much clearer.
> > > >
> > > > We hope that these clarifications reinforce the reviewer’s confidence in the value and novelty of the work, and we greatly appreciate their openness to update their score.

---

### Official Review · Reviewer_oY2q · 2025-11-01

**Soundness:** 3
**Presentation:** 3
**Contribution:** 3
**Rating:** 6
**Confidence:** 3

**Summary:**

This paper introduces novel generalization bounds which depend on the regularity of the sampling distribution rather than that of the model. By recombining ideas (local partitioning of the input space plus concentration for multinomial/binomial counts) into a new model‑ and data‑dependent bound that you can evaluate from the training set alone, without compression, and works on Imagenet scale. That said, the bound is still loose (often ~2–3× the true test error) and depends on a chosen partition, so it is more a pragmatic workaround than a final theory of generalization. There is a lot of hard work that goes into getting the numerical bound on the actual NN tighter, without bounding a compressed NN, as in https://arxiv.org/abs/2211.13609  which is a different object

The suggested convergence rate O(n−1/4) leaves plenty room to improve, and the bound is also sensitive to the partition K

**Strengths:**

The paper brings an original contribution to what is a huge literature on PAC-Bayes bounds.  There is a good literature review and an extensive test on empirical data.

**Weaknesses:**

The tight bounds generate less insight than looser bounds can.
 The major draw back is that, just like  compression bounds of  https://arxiv.org/abs/2211.13609, the bound is mostly collapsed to training error. And it's much worse than the compression bound in terms of tightness. Presumably that's the price you need to pay for developing a theory that bounds the actual network.

**Questions:**

Q1 - how does this work relate to prior non-vacuous bounds such as the marginal-likelihood one in https://arxiv.org/abs/2012.04115

Q2 - in your main bound - can you give an intuitive description of each term, and explain where it may break down, e.g. under parameter re-scalings?

---

> ### Author Response · Authors · 2025-11-20
> **Response to the questions**
>
> We thank the reviewer for the thoughtful assessment and constructive criticism. We address  each question and concern below and clarify the contributions, tightness, and novelty of our results.
>
> ### 1. The questions
>
> > Q1: How does the work relate to non-vacuous marginal-likelihood bounds such as the one in _Valle-Pérez & Louis (2020)_ (arXiv:2012.04115)?
>
> Valle-Pérez & Louis (2020) develop a *marginal-likelihood PAC-Bayes* viewpoint and show (in their Theorem 5.1) that, for any $\nu,\eta>0$,  the following holds with probability at least $1-\nu$ over the sampling of $S$:
>
> $\Pr_h\left[ F(P,h) < 1- \exp\left(-\frac{1}{n-1} [-\ln(Q(S)) + \ln n - \ln\nu - \ln\eta] \right) \right] \ge 1- \eta$
>
> where $Q$ is the distribution on a hypothesis space $\mathcal{H}$, $Q(S) = \sum_{h \in C(S)} Q(h)$ corresponds to the marginal likelihood, or Bayesian evidence of the data $S$, $C(S)$ is the set of hypotheses in $\mathcal{H}$ consistent with the sample $S$.
>
> Meanwhile we show the following holds with probability at least $1- \gamma^{-\alpha}  -\delta$ over the sampling of $S$:
>
> $F(P, h) \le F(S,h)  + C\sqrt{\hat{u}\alpha \ln\gamma } + g_2(\delta/2)$
>
> Our approach differs in both spirit and application:
> - The marginal-likelihood bound reasons about distribution over models, while our bound is a direct, deterministic guarantee for a _specific/trained model_.
> - The marginal-likelihood bound cannot be exactly computed and relies on approximations to evaluate $Q(S)$, while ours is exactly computable from the training set without constructing a posterior over predictor.
>
> In short, Valle-Pérez & Louis provide a powerful PAC-Bayes lens that helps explain when and why PAC-Bayes can be tight; our contribution is complementary — it avoids the posterior/KL machinery altogether and produces an explicit, computable certificate for the _deployed_ model, trading some tightness for direct applicability at ImageNet scale and 600M-parameter models.
>
> Thus, our approach is fundamentally different in methodology, conceptual motivation, and application scope.
>
> > Q2: Can you give intuition for each term and explain potential failure modes?
>
> We appreciate this request and have strengthened the intuition in the revision. The structure is:
>
> $F(P,h) \le F(S,h) + C \sqrt{\frac{u}{2 n^2} \ln(1/\delta_1)} + g(\Gamma,h,\delta_2)$
>
> -   $F(S,h)$: empirical loss.
>
> -   **Term 1 (distribution complexity)**:
>
>     -  $u$ contains $\sum_i p_i^2$, the _local concentration_ of the data distribution under the partition. It also encodes the complexity of the data distribution.
>     -   If the data distribution is smoother or spreads uniformly across the partition, the term is small. Otherwise this term can be high, suggesting a misalignment between the partition and data distribution.
> -   **Term 2 (local model behavior)**: $g(\Gamma,h,\delta_2)$
>     -   Encodes how the model behaves inside each region $Z_i$.
>     -   If the model’s local loss is small in regions with many samples, $g$ becomes small.
>     -   This is what makes the bound _model-dependent_ rather than architecture-dependent.
>
> **Potential failure modes:**  The bound may be looser when
> 1.  The partition is misaligned with data geometry, i.e. $P(Z_i) \ll P(Z_j)$ for some $i, j$.
> 2. The sample size $n$ is small. In this case, the binomial counts $n_i$ may not approximate $n p_i$ well, leading to a large value for **Term 1** before.
> 3. The trained model underfits, leading to a large empirical loss.
>
> We thank the reviewer for encouraging us to clarify these points which substantially improve the paper.

---

> > ### Author Response · Authors · 2025-11-20
> > **Response to other concerns**
> >
> > ### 2. Other comments
> >
> > > C1: The bound is still loose (≈2–3× test error), depends on a chosen partition, and behaves similarly to compression bounds by being close to the empirical error.
> >
> > We acknowledge that the current bound is not as tight as the best hand-crafted compression-based PAC-Bayes bounds. However, the comparison is not like-for-like: **our bound certifies the _actual_ trained model**, without _any_ compression, quantization, finetuning. This is a fundamental difference.
> >
> > Compression-based bounds achieve tightness precisely because the model is aggressively simplified until its complexity matches the bound's requirements. But the certified guarantee is for the _compressed_ model $h'$, not the original trained network $h$. As discussed in the paper, and empirically demonstrated in prior work, $h$ and $h'$ may differ significantly in behavior. Thus, these bounds do not give a theoretical guarantee for the network deployed in practice.
> >
> > **Our contribution is orthogonal:**
> >
> > -   We introduce a *new form of model-dependent generalization bound* where the complexity arises from the _data distribution_ and the _local behavior_ of the model, not from network architecture or parameter norms.
> > -   The bound is *exactly computable from the training set alone*, at ImageNet scale, for deep architectures of up to *600M parameters*.
> > -   No prior theory can currently give a non-vacuous guarantee for the _unmodified_ trained ImageNet models we evaluate.
> >
> > Thus, even at 2–3× the test error, the bound is meaningful: it remains **non-vacuous in a setting where all existing generalization frameworks are impractical or vacuous**, except after heavily modifying the model.
> >
> > Finally, we emphasize that the “2–3×” factor is achieved with _non-optimized parameters_ and a _generic partition_. As shown in our ablations in Appendix D, the gap can be significantly reduced through better data-aligned partitions.
> >
> > >C2: The tight bounds generate less insight than looser bounds can. The major drawback is that, just like compression bounds, the bound is mostly collapsed to training error.
> >
> > We agree that many non-vacuous bounds—including compression-based PAC-Bayes bounds—end up numerically close to the empirical error. However, this phenomenon arises for fundamentally different reasons in our framework, and it reflects a _strength rather than a limitation_ of the proposed approach.
> >
> > Compression bounds collapse toward the training error because the compressed model is heavily simplified until its effective capacity matches the size of the training set; the resulting guarantee pertains to a _different model_ and provides limited insight into why the _original_ network generalizes.
> >
> > In contrast, our bound approaches the empirical error because the empirical loss is **exactly the correct leading term** when the model behaves consistently across the partition regions and the data distribution is sufficiently regular. The gap in our bound is governed by _observable, interpretable quantities_: the local concentration of the data distribution $(\sum p_i^2)$ and the model’s local average losses $(a_i(h))$. When these are small, the true error is provably close to the training error. Thus, the “collapse” is not an artifact of forcing the model into a simpler representation, but an indication that the model’s behavior is locally stable with respect to the data geometry.
> >
> > This leads to a different kind of insight: our method highlights _which_ regions of the input space and _which_ local behaviors of the trained network are responsible for the generalization gap. Unlike compression bounds, this decomposition is directly tied to the _unmodified_ network, and our ablations (Appendix D) show that the bound sharpens when partitions align with the intrinsic data geometry. In this sense, the bound does not obscure insight; rather, it shifts the explanatory burden from parameter-space compressibility to **data-dependent local regularity**, a property that seems more faithful to how large networks behave in practice.

---

### Official Review · Reviewer_kh9d · 2025-11-04

**Soundness:** 2
**Presentation:** 1
**Contribution:** 3
**Rating:** 6
**Confidence:** 3

**Summary:**

This paper revisits the classical generalization problem in deep learning and proposes non-vacuous generalization bounds for models that are trained without being compressed or optimized to make the bounds tighter. The authors derive new bounds that depend on the complexity of the data distribution and the local behavior of the hypothesis in different areas of the data space. They also propose a version of this bound that can be computed in practice using the training data and without any changes to the pretrained model. The authors demonstrate that these bounds are non-vacuous for models pretrained on ImageNet, proving the first non-vacuous bounds for non-altered models at the scale of ImageNet.

**Strengths:**

- The paper addresses a long-standing criticism that generalization bounds for deep networks are constructed for altered versions of the models and do not apply to the original models, or models used in practice.
- The derivation of the bound and its empirical computation are both sound.
- The bounds are non-vacuous in practice and the gap between the bounds and the test error is not too big compared to even sota bounds.

**Weaknesses:**

- The transition between the bounds is very poorly explained, or not explained at all such as the case for 3.2. While including a proof sketch is not required, it would be at least important to keep the flow of the work by explaining why we needed the first two bounds to get to the third bounds and how they connect and differ.
- Overall, the presentation of the notation and theorems is super hard to follow and took a very long time to understand qualitatively what the different quantities measure. More clarity is definitely required.
- Last point on the presentation: overall the quality of the writing can be improved to be more concise as it takes until the end of page 4 to get to the meat of the paper. The first paragraph of the introduction for instance can be easily removed.
- While the gap between the test error and the bound is not as large as some of the classical bounds, it is still not totally tight and therefore it would have been interesting to see if the bounds track behavior we see in practice, e.g., CNNs being more compressible and generalizing better than MLPs given the same number of parameters.
- It is unclear how the optimization over the hyperparameters is accounted for in the bounds to keep them valid.
- While the lack of baselines is understandable given that other approaches do require changing the model, it is worth noting that PAC-Bayes bounds can be computed with a collapsed posterior around the solution found and do not need to be computed for a stochastic model.
- The evaluation is limited to ImageNet only; and while this is indeed a very challenging dataset, it would be interesting to see how tight the bounds are for other datasets.

**Questions:**

- If you optimize your bounds over hyperparameters, do you pay a union bound penalty to make the bound simultaneously hold over all hyperparameters like the authors do in "PAC-Bayes Compression Bounds So Tight That They Can Explain Generalization ", https://arxiv.org/abs/2211.13609?
-  Can you compute baseline bounds proposed in previous works without changing the model at all? If the bounds are vacuous, that would serve as a confirmation that your work indeed improves upon existing works. E.g., for compression bounds that use linear subspace projection, you could try to inject the KL without any subspace projection, ie in the original space, and see if the bounds are indeed vacuous.
- Can you compute the correlation coefficient between the bounds and the test error? It would be interesting to see to which extent the bounds do track the test error.
- Can you use the bounds to provide better understanding of generalization in practice. eg, why SwinTransformers perform better than ResNets?

---

> ### Author Response · Authors · 2025-11-20
> **Response to the questions**
>
> We thank the reviewer for carefully reading the manuscript and for the constructive comments on presentation and clarity. The manuscript has been substantially improved in structure, clarity, and readability in direct response to these comments. Below we address the questions first and then the remaining concerns.
>
> ### 1. Main questions
>
> >**Q1**: If you optimize your bounds over hyperparameters, do you pay a union bound penalty to make the bound simultaneously hold over all hyperparameters?
>
> We'd like to remind that the confidence in Theorem 3.2 is $1-\gamma^{-\alpha} -\delta$. In our experiments, we chose $\gamma = 0.04^{-1/\alpha}$ and hence $\gamma^{-\alpha} = 0.04$ for any given $\alpha$. Therefore, with $\delta=0.01$, the confidence is always $1-\gamma^{-\alpha} -\delta = 0.95$.
>
> For such choices for $(\gamma,\delta)$, Theorem 3.2 basically shows:
>
> $\Pr(A(K,\alpha)) \ge 0.95$
>
> for all $K$ and $\alpha$ satisfying the mild condition of the theorem, where $A$ denotes an event that depends on $K$ and $\alpha$. Since our optimization for $K$ and $\alpha$ satisfying the theorem's condition, the optimized values $K'$ and $\alpha'$ should also ensure $\Pr(A(K',\alpha')) \ge 0.95$. So no union bound is required.
>
> >**Q2**: Can you compute baseline bounds proposed in previous works without changing the model at all? If the bounds are vacuous, that would serve as a confirmation that your work indeed improves upon existing works. E.g., for compression bounds that use linear subspace projection, you could try to inject the KL without any subspace projection, ie in the original space, and see if the bounds are indeed vacuous.
>
> Thank you for this helpful suggestion. In **Appendix D.2**, we already compute numerical baselines for two existing bounds. These prior bounds cannot be evaluated exactly from the training set alone because several terms (e.g., the local error $a_i$) are not observable. Following the standard practice in these works, we use a small validation set to approximate those quantities. Under this setting, one baseline is occasionally non-vacuous while the others are fully vacuous. Importantly, our bound remains strictly tighter **without requiring any external data**, which already demonstrates the improvement you requested.
>
> Regarding the reviewer’s suggestion to “inject the KL term without performing compression,” we would like to clarify that such a comparison is **not well-posed**. PAC-Bayes and compression bounds yield guarantees for a _posterior distribution over models_ (or a compressed surrogate), i.e. bounding $E_{g}[F(P,g)]$. This quantity is not guaranteed to upper-bound the true error of a specific/trained model $h$. Injecting a KL term in the original parameter space, without constructing a valid posterior or compression operator, breaks the assumptions required for PAC-Bayes guarantees and may not produce a meaningful or comparable bound. Thus, such an experiment would not be an informative baseline.
>
> Our experimental protocol instead focuses on bounds that target the **same object** as ours: the expected error $F(P,h)$ of the _actual trained model_, without modifying it. This choice ensures a fair comparison. Across 32 modern architectures—including ImageNet-scale pretrained models with >600M parameters—our bound is the only one that remains both **computable** and **non-vacuous** without altering the model or invoking auxiliary datasets. To the best of our knowledge, no prior work has achieved certificates of this scale for practical, unmodified networks.
>
> We clarify this point in lines 355-363, and we appreciate the reviewer’s suggestion.
>
> >**Q3**: Can you compute the correlation coefficient between the bounds and the test error? It would be interesting to see to which extent the bounds do track the test error.
>
> We have already provided Table 5 in Appendix D.3. The results show that our bounds have high correlations to the test error. Such correlations even can be increased when computing those bound from a validation set.

---

> > ### Author Response · Authors · 2025-11-20
> > **Response to other comments (continue)**
> >
> > >**Q4**: Can you use the bounds to provide better understanding of generalization in practice?
> >
> > Thank you for this insightful suggestion. In order to explore deeper behaviors of a model $h$, we can focus on the following quantities in our Bound (1):
> > - $Align(h) = \sum_{i \in T} a_i\sqrt{n_i/n}$: It tells how well the model's error can match with the data distribution. A better model should align better with the distribution's complexity, hence making $Align$ smaller.
> > - $Fair(h) = \sum_{i \in T} a_i$: It tells the macro-level error of model $h$. It also suggests how fair on average is the model for different local areas.
> > - $Behavior(h) = Align(h)* \sqrt{2\ln(2K/\delta_2)/n} + Fair(h)*2\ln(2K/\delta_2)/n$: the unified behavior, being an important part of Bound (1).
> >
> > We use $K=200, \delta_2=0.01$ as in the main paper and the ImageNet validation set to compute those quantities. The results for 6 pretrained models are reported below.
> >
> > | model | Test error | $Align$ | $Fair$ | $Behavior$ |
> > | ----- | --- | --- | --- | --- |
> > | ResNet18 V1 |  0.30242 | 3.28869  | 64.6831   | 0.01444  |
> > | ResNet101 V1 | 0.22626 | 2.43343  | 46.0864   | 0.01066  |
> > | ResNet152 V2 | 0.17716 | 1.93093  | 38.8202   | 0.00850 |
> > | DenseNet201 |  0.23104 | 2.38285  | 45.8748   | 0.01045  |
> > | SwinTransformer B | 0.15888 | 1.72302  | 33.1667   | 0.00756 |
> > | VIT B 16 linear | 0.18114 | 2.14093  | 43.7663   | 0.00943  |
> > | **Correlation to test error** | | **0.98582** |  **0.96397** | **0.98468** |
> >
> > We can observe that all of those quantities have extremely high correlations to the test error. $Align$ has the highest correlation, but $Fair$ has the lowest one. These results demonstrate that **$Align$ can exhibit the quality of a model, and can be an accurate indicator for comparison between two models.** When visualize the two lists {$a_i: i \in T$} and {$\sqrt{n_i/n}: i \in T$}, we observe a near perfect correlation for some models e.g. SwinTransformer B. Meanwhile ResNet18 V1 exhibited a much worse correlation.
> >
> > We added Appendix D.4 in the revised version of the paper to summarize this analysis.
> >
> > ### 2. Other comments
> >
> > > **C1**: On the transitions between the bounds
> >
> > We appreciate this feedback and agree that the earlier version did not make the progression between the three bounds sufficiently explicit. In the revised manuscript, we now provide a clear narrative explaining the role of each bound:
> >
> > -   Theorem 3.1 establishes a general upper bound using the exact (but intractable) terms $u$ and $g$. It serves as the theoretical foundation.
> >
> > -   Theorem 3.2 removes the distribution-dependent components by introducing directly computable surrogates $\hat{u}$ and $g_2$. This is the crucial bridge from theory to practice.
> >
> > We highlight (using blue color) these motivations explicitly in the introduction to Section 3 and in the paragraphs preceding and after Theorem 3.2. The flow is now clearly stated: _Theorem 3.1 provides the conceptual structure → Theorem 3.2 approximates the intractable terms, leading to a fully computable bound._
> >
> > We thank the reviewer for pushing us to make these dependencies clearer.
> >
> > > **C2**:  On notation and interpretability of quantities
> >
> > We acknowledge that the first submission contained dense notation and insufficient explanations of what certain terms intuitively measure. This has now been significantly revised:
> >
> > -   A dedicated *notation summary* is added at the start of Section 3.
> > -   Each main quantity (e.g. $u, \hat{u}, g, g_2$) is followed by some plain-language sentences explaining its role.
> > -   Transitions such as “why this term appears” and “what it controls” have been added to avoid abrupt introduction of symbols.
> >
> > These changes substantially improve readability and should eliminate the confusion noted by the reviewer.
> >
> > > **C3**:  On conciseness and the introduction
> >
> > We appreciate the comment regarding pacing. While we summarize several lines of prior work to properly position our contribution within the broad literature on neural-network generalization, we have now streamlined this discussion to improve conciseness. The revised introduction presents the necessary background more efficiently and leads into the main technical contributions in Section 3 earlier.
> >
> > > **C4**: On the remaining gap between test error and the bound
> >
> > We fully agree that the bound is not perfectly tight. Our goal is not to claim a near-equality but to *significantly reduce the gap compared to prior computable bounds*, where most existing approaches become impractical or vacuous. The experiments in Section 5  and Appendix D demonstrate that our bounds allows us to produce the first non-vacuous certificates for ImageNet-scale pretrained models exceeding 600M parameters _without altering them_. To the best of our knowledge, this capability is unique.
> >
> > We clarify in the conclusion that while our bounds are promising, perfect tightness is not expected for high-dimensional deterministic bounds and remains an open challenge.

---

### Author Response · Authors · 2025-11-28
**An inquiry**

Dear Reviewers, ACs, and PCs,

We would like to express our sincere appreciation for the detailed evaluations and constructive comments provided throughout the review process. We have revised the manuscript accordingly, addressing every point raised—clarifying the theoretical framing, strengthening the intuition behind the bounds, incorporating additional analyses, and improving the overall presentation. Reviewer x2iF has noted that these revisions substantially enhance the clarity and impact of the work.

In light of these updates, we respectfully inquire whether reviewers (particularly **JrsG** and **Ygiq**) might consider revisiting their assessments. Both reviewers identified meaningful strengths in the original submission, and we hope the revisions resolve the concerns surrounding clarity, positioning, and the interpretation of empirical results. We believe the refined presentation now more clearly reflects the novelty and significance of the proposed framework.

We appreciate the considerable time and expertise each reviewer has invested in evaluating this work, and we remain grateful for any further thoughts or updated assessments the reviewers may wish to provide.

Warm regards,

The Authors

---

### Meta-Review · Area_Chair_JbRX · 2026-01-03

**Summary:**

This paper studies non-vacuous and computable generalization bounds for large-scale deep neural networks. Two reviewers (kh9d and oY2q) recommend acceptance, while three reviewers (x2iF, JrsG, and Ygiq) recommend rejection. The main concerns focus on the quality of the presentation, whether the results provide genuinely new insight into generalization, the relationship and comparison to existing bounds (especially PAC-Bayes), and the numerical tightness of the proposed bounds.

In the rebuttal, the authors made efforts to improve the presentation and to clarify several points. However, in response to Reviewer x2iF, the added experiments include classical models (e.g., logistic regression and SVM), where the proposed bound seems noticeably looser than standard learning theory bounds. The authors argue that this may stem from limited training data and reflects the data-dependent nature of their bound. Still, this looseness in the small-to-moderate data regime is a meaningful drawback. In particular, it raises a key practical question, namely for what regimes of sample size does the bound become meaningfully tight? I would strongly prefer that the authors not stop at stating their result is merely "complementary" to classical bounds without further investigation. Related theoretical work on whether uniformly tight generalization bounds can even exist at all (see [R1]) may provide useful guidance for a deeper follow-up.

While I agree that presentation issues alone are typically not decisive, the more central question here is whether the bounds meaningfully advance our understanding of generalization, beyond simply being non-vacuous in some regimes. Moreover, in the broader deep learning generalization literature, it is indeed common for numerically tighter bounds to provide limited algorithmic insight, while looser (even vacuous) bounds can sometimes be more conceptually informative (e.g., norm-based analyses). For example, mutual information-based bounds have achieved extremely tight bounds for deep networks (see [R2–R5]), yet their conceptual takeaways can be debated. A related point is that the claim that "computing MI in high-dimensional, non-linear settings is computationally challenging" (Lines 145–146) seems overstated in light of [R2–R5], which provide practical ways to estimate MI-based bounds (often without altering the trained model, though they may require a ghost dataset). Taken together, this raises an important motivation question, what is the intended value of numerical tightness if it does not translate into clearer insight or actionable understanding of generalization? I hope the authors address this more directly in the next revision.

[R1] Michael Gastpar, et al. Fantastic Generalization Measures are Nowhere to be Found. ICLR 2024.

[R2] Hrayr Harutyunyan, et al. Information-theoretic Generalization Bounds for Black-box Learning Algorithms. NeurIPS 2021.

[R3] Fredrik Hellström, et al. A New Family of Generalization Bounds using Samplewise Evaluated CMI. NeurIPS 2022.

[R4] Ziqiao Wang, et al. Tighter Information-theoretic Generalization Bounds from Supersamples. ICML 2023.

[R5] Yuxin Dong, et al. Exactly Tight Information-theoretic Generalization Bounds via Binary Jensen-Shannon Divergence. ICML 2025.

**Reviewer Concerns:**

Reviewer kh9d's concerns about presentation and several clarification points seem to have been addressed by the authors.

Reviewer Ygiq requests experiments on language models. The authors responded that this setting does not satisfy the i.i.d. assumption required for applying their proposed bounds, which is a reasonable explanation.

Several reviewers question whether the bounds provide genuinely novel insights into generalization. In my view, the authors have partially addressed this, but the point likely still warrants further discussion, as noted above.

Reviewer JrsG criticizes the paper's presentation, technical novelty, and missing key references. The authors have taken these comments into account and revised the manuscript accordingly. While some concerns have been addressed (and I agree that a strong rejection may be harsh), I suspect the reviewer may remain unconvinced on certain points. For example, the claim that "PAC-Bayes provides guarantees for an averaged model and typically requires compression or quantizationD" may not fully satisfy this reviewer, given that there exist PAC-Bayes bounds for deterministic models (e.g., the paper "A General Framework for the Practical Disintegration of PAC-Bayesian Bounds", which is cited in the submission).

**Reviewer Scores:**

Reviewer kh9d and Reviewer oY2q will likely maintain their positive scores. While some concerns have been addressed in the revision, there is little indication from their initial comments that they intend to actively champion the paper beyond their current stance.

Reviewer x2iF may increase the score, as the reviewer explicitly expressed openness to updating it. That said, this reviewer also highlighted more fundamental limitations of the current submission.

Reviewer JrsG, who assigned a score of 0 in the initial review, may revise upward given the improvements in presentation. However, I expect the updated score would still lean toward rejection.

It is unclear whether Reviewer Ygiq will adjust their score. It is quite possible the reviewer will continue to request a more thorough comparison between the proposed bounds and previous work, since the original request for such comparisons has not been fully satisfied.

Overall, the final score may be quite borderline.

---

### Decision · Program_Chairs · 2026-01-26

Reject